# PIE-1 SUMOylation promotes germline fates and piRNA-dependent silencing in *C. elegans*

**Heesun Kim[1][†], Yue-He Ding[1][†], Shan Lu[2], Mei-Qing Zuo[2], Wendy Tan[1], Darryl Conte Jr[1], Meng-Qiu Dong[2], Craig C Mello[1,3]***

[1]RNA Therapeutics Institute, University of Massachusetts Medical School, Worcester, United States; [2]National Institute of Biological Sciences, Beijing, China; [3]Howard Hughes Medical Institute, Chevy Chase, United States

**Abstract** Germlines shape and balance heredity, integrating and regulating information from both parental and foreign sources. Insights into how germlines handle information have come from the study of factors that specify or maintain the germline fate. In early *Caenorhabditis elegans* embryos, the CCCH zinc finger protein PIE-1 localizes to the germline where it prevents somatic differentiation programs. Here, we show that PIE-1 also functions in the meiotic ovary where it becomes SUMOylated and engages the small ubiquitin-like modifier (SUMO)-conjugating machinery. Using whole-SUMO-proteome mass spectrometry, we identify HDAC SUMOylation as a target of PIE-1. Our analyses of genetic interactions between *pie-1* and SUMO pathway mutants suggest that PIE-1 engages the SUMO machinery both to preserve the germline fate in the embryo and to promote Argonaute-mediated surveillance in the adult germline.

**\*For correspondence:**
Craig.Mello@umassmed.edu

[†]These authors contributed equally to this work

**Competing interests:** The authors declare that no competing interests exist.

## Introduction

During every life cycle, the eukaryotic germline orchestrates a remarkable set of informational tasks that shape heredity and create variation necessary for the evolution of new species. One approach for understanding the mechanisms that promote germline specification and function has been the identification of genes whose protein products localize exclusively to the germline and for which loss-of-function mutations result in absent or non-functional germ cells and gametes (*Seydoux and Braun, 2006*). In *Caenorhabditis elegans*, PIE-1 is a key regulator of germline specification (*Mello et al., 1992*). The *C. elegans* zygote, P0, undergoes a series of asymmetric divisions that generate four somatic founder cells and the germline blastomere P4. The PIE-1 protein is maternally deposited and uniformly present in the cytoplasm and nucleus of the zygote, but rapidly disappears in each somatic blastomere shortly after division (*Mello et al., 1996*; *Reese et al., 2000*; *Tenenhaus et al., 1998*). In *pie-1* mutants, the germline lineage differentiates into extra intestinal cells causing an embryonic arrest (*Mello et al., 1992*). PIE-1 localizes prominently in nuclei in the early P-lineage blastomeres and persists in the primordial embryonic germ cells through much of embryogenesis (*Mello et al., 1996*). The presence of PIE-1 correlates with global hypo-phosphorylation of the C-terminal domain (CTD) of RNA polymerase II (pol II) in germline blastomeres (*Seydoux and Dunn, 1997*), and some studies suggest that PIE-1 may directly inhibit the CTD kinase to prevent transcriptional activation (*Batchelder et al., 1999*; *Ghosh and Seydoux, 2008*).

PIE-1 is a member of the tandem CCCH zinc finger protein family (*Blackshear et al., 2005*). PIE-1 differs from most of its homologs in having a prominent nuclear localization. However, like the majority of its family members, PIE-1 also localizes in the cytoplasm where it is thought to bind and regulate the expression of germline mRNAs, including the *nos-2* mRNA (*Tenenhaus et al., 2001*). Hints at the nuclear function of PIE-1 came from a yeast two-hybrid screen which identified the

Krüppel-type zinc finger protein MEP-1 as a PIE-1 interacting factor (*Unhavaithaya et al., 2002*). MEP-1 co-purifies with LET-418, a homolog of mammalian ATP-dependent nucleosome remodeling factor Mi-2 (*von Zelewsky et al., 2000*), and with HDA-1, a homolog of mammalian histone deacetylase HDAC1 (*Shi and Mello, 1998*). Inactivation of maternal *mep-1* and *let-418* causes a striking developmental arrest of L1-stage larvae, whose somatic cells adopt germline-specific transcriptional programs, and assemble germline-specific peri-nuclear nuage-like structures called P granules (*Unhavaithaya et al., 2002*). These soma-to-germline transformations depend on the trithorax-related protein MES-4 and components of a polycomb repressive complex (PRC2) (MES-2 and MES-3) (*Unhavaithaya et al., 2002*), whose functions are thought to promote fertility by maintaining germline chromatin (*Strome and Updike, 2015*). Taken together, these previous studies on PIE-1 suggest that it functions as a master-regulator of the germline fate in *C. elegans* embryos, preventing somatic differentiation, while also protecting the germline chromatin from remodeling. However, the possible biochemical mechanisms through which this small CCCH zinc finger protein exerts its dual effects on transcription and chromatin in the germline were entirely unknown.

Here, we show PIE-1 promotes the regulation of its targets at least in part through the small ubiquitin-like modifier (SUMO). By yeast two-hybrid screening, we show that PIE-1 engages the highly conserved E2 SUMO ligase UBC-9. UBC-9 enzymes catalyze the addition SUMO to lysine residues on target proteins (*Capili and Lima, 2007b*; *Geiss-Friedlander and Melchior, 2007*; *Johnson, 2004*). The reversible addition of SUMO (or SUMOylation) and its removal by de-SUMOylating enzymes is thought to occur on thousands of substrate proteins with diverse functions, especially nuclear functions including DNA replication, chromatin silencing, and the DNA damage response (*Hendriks and Vertegaal, 2016*). SUMOylation can have multiple effects and is not primarily associated with the turnover of its targets, but rather is often associated with changes in protein interactions. For example, SUMOylation of a protein can promote interactions with proteins that contain SUMO-interacting motifs (*Matunis et al., 2006*; *Psakhye and Jentsch, 2012*; *Shen et al., 2006*).

Through a series of genetic and biochemical studies, we show that the SUMO pathway promotes the activity of PIE-1 in preserving the embryonic germline. We show that PIE-1 is itself modified by SUMO. Paradoxically, PIE-1 is not SUMOylated in early embryos, but rather in adult animals where PIE-1 was not previously known to be expressed or functional. Indeed, CRISPR-mediated GFP tagging of the endogenous *pie-1* locus confirmed uniform nuclear expression of PIE-1 protein throughout the meiotic zone and in oocytes of adult hermaphrodites. Using whole proteome analysis for detecting SUMO-conjugated proteins, we identify the type 1 HDAC, HDA-1, as a protein modified by SUMO in a PIE-1-dependent manner. Surprisingly, whereas PIE-1 was originally thought to inhibit the MEP-1/Mi-2/HDA-1 complex in embryos, we show that in the germline PIE-1 acts in concert with the SUMO pathway to promote the association of MEP-1 with HDA-1, and to maintain the hypoacetylation of germline chromatin. Although *pie-1* lysine 68 mutants that prevent PIE-1 SUMOylation are viable, we show that they exhibit synthetic lethality in combination with null alleles of *gei-17*, a *Drosophila Su(var)2–10* SUMO-E3 ligase homolog (*Hari et al., 2001*; *Mohr and Boswell, 1999*; *Ninova et al., 2020*), and that double mutants exhibit potent desilencing of a piRNA Argonaute sensor. Our findings are consistent with a model in which PIE-1 engages SUMO to preserve the embryonic germline fate, and also to promote the assembly of a MEP-1/Mi-2/HDA-1 chromatin remodeling complex required for inherited Argonaute-mediated gene silencing in the adult hermaphrodite germline.

## Results

### PIE-1 is SUMOylated in adult germ cells

To explore how PIE-1 promotes germline specification, we sought to identify protein interactors. Immunoprecipitation of PIE-1 protein from embryo extracts proved to be challenging because the protein is expressed transiently in early embryos, where it is only present in early germline cells. Moreover, PIE-1 was insoluble and unstable in worm lysates preventing the analysis of PIE-1 complexes by immunoprecipitation (*Figure 1—figure supplement 1*). We therefore performed a yeast two-hybrid screen to identify PIE-1 interactors (*Figure 1A* and *Supplementary file 1*; see Materials and methods). As expected, this screen identified the Krüppel-type zinc finger protein MEP-1, a known PIE-1 interactor and co-factor of the Mi-2/NuRD (nucleosome remodeling deacetylase)

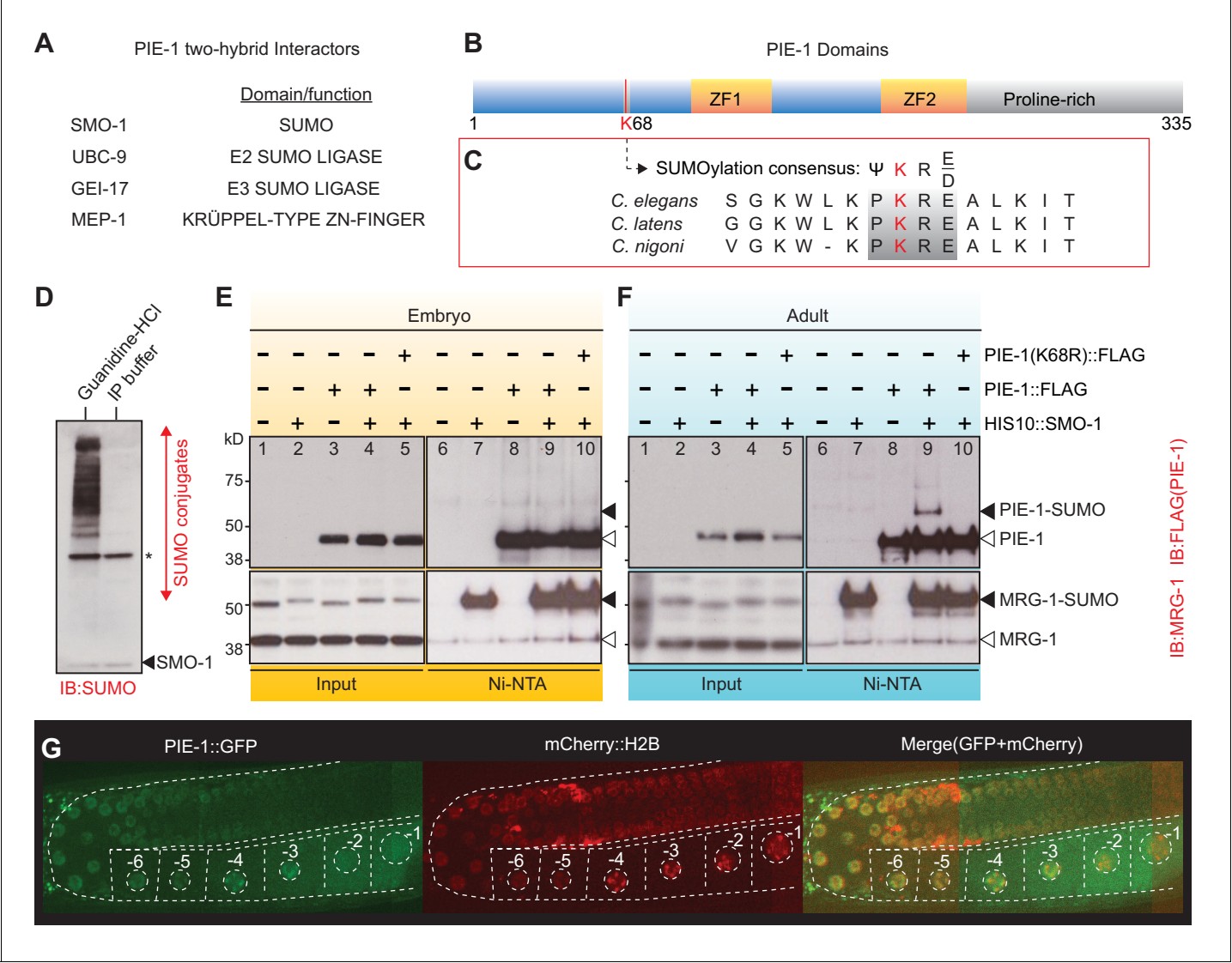

**Figure 1.** PIE-1 is SUMOylated on K68 residue in the *Caenorhabditis elegans* germline. (**A**) Summary of PIE-1 interactors identified by yeast two-hybrid screen (see *Supplementary file 1* for complete list). (**B and C**) Domain structure of PIE-1 containing two zinc fingers (ZF1 and ZF2) and proline-rich region, and location (red bar) of a consensus small ubiquitin-like modifier (SUMO) acceptor motif (ψKXE, where ψ represents a hydrophobic amino acid, K is the acceptor lysine, and X is any amino acid) conserved in PIE-1 from other *Caenorhabditis* species. (**D**) Western blot analysis of SUMO-conjugated proteins in total worm lysates prepared with guanidine-HCl denaturing buffer or IP buffer. The resistant band (asterisk) migrates with the expected size of the E1 enzyme AOS-1, which attaches to SUMO by a thioester bond and may therefore resist SUMO proteases, which cleave isopeptide bonds. The black triangle indicates free SMO-1. (**E and F**) Western blot analyses of SUMOylated proteins enriched from (**E**) early embryo or (**F**) adult lysates from wild-type *pie-1::flag* or *pie-1(K68R)::flag* worms. SUMOylated proteins were enriched from worms expressing HIS10::SMO-1 by Ni-NTA chromatography. Black triangles indicate SUMOylated forms of PIE-1 or MRG-1. White triangles indicate unmodified PIE-1 or MRG-1. MRG-1 is a robustly SUMOylated protein (*Supplementary files 2* and *3*; *Drabikowski et al., 2018*; *Kaminsky et al., 2009*) and thus serves as a positive control. (**G**) Confocal images of PIE-1::GFP and mCherry::H2B in adult germline of live *pie-1::gfp; pie-1p::mCherry::his-58* worms. Oocyte nuclei are indicated by white circles and numbered.

The online version of this article includes the following figure supplement(s) for figure 1:

**Figure supplement 1.** PIE-1 is insoluble and unstable.

**Figure supplement 2.** Enrichment of SUMOylated proteins from worms expressing HIS-tagged SMO-1.

**Figure supplement 3.** PIE-1 expression in the adult germline and early embryos.

complex (*Passannante et al., 2010*; *Unhavaithaya et al., 2002*). In addition, this screen identified the small ubiquitin-like modifier SMO-1 (SUMO); the E2 SUMO-conjugating enzyme UBC-9 (*Jones et al., 2002*); and GEI-17 (*Holway et al., 2005*; *Kim and Michael, 2008*), a homolog of vertebrate PIAS2 and *Drosophila* Su(var)2–10, an E3 SUMO ligase (*Ninova et al., 2020*; *Pichler et al., 2017*).

Motif analysis predicted one consensus SUMO acceptor site (ψKXE; *Rodriguez et al., 2001*) in PIE-1 that is perfectly conserved in PIE-1 orthologs of other *Caenorhabditis* species (*Figure 1B and C*). Although covalent in nature, the addition of SUMO to substrates is rapidly reversible by SUMO protease enzymes that are active under conditions typically used to prepare lysates for immunoprecipitation studies (*Di Bacco et al., 2006*; *Zhang et al., 2017*). For example, even the preparation and centrifugation of lysates at 4°C followed by reconstitution in IP buffer with no other incubation resulted in the complete removal of SUMO from its target proteins (*Figure 1D*). To identify SUMO-conjugated proteins, we inserted a poly-histidine epitope into the endogenous *smo-1* gene, and then used nickel (Ni) affinity chromatography under stringent denaturing conditions to enrich SUMOylated proteins (*Tatham et al., 2009*). Eluates from Ni-affinity chromatography were then analyzed by mass spectrometry (MS) and western blotting (*Figure 1—figure supplement 2B and C* and see below).

In principle, it is possible to directly detect SUMOylated peptides by mass spectrometry (*Impens et al., 2014*; *Knuesel et al., 2005*). To create a cleavage signature suitable for high-throughput identification of modified peptide fragments, leucine 88 of SUMO must be substituted with lysine. Unfortunately, editing the endogenous *smo-1* locus to encode the L88K mutation caused a lethal *smo-1* phenotype in *C. elegans* (data not shown).

Previous proteomic analyses of SUMOylated worm proteins have relied on *smo-1* transgenes that encode HIS-tagged SUMO fused to the FLAG epitope or to GFP (*Drabikowski et al., 2018*; *Kaminsky et al., 2009*). Inserting in-frame FLAG or mCherry sequences into the endogenous *smo-1* locus caused pronounced hypomorphic *smo-1* phenotypes (data not shown). We therefore inserted hexahistidine (HIS6) or decahistidine (HIS10) sequences directly after the initiator methionine codon of *smo-1* without any other sequences. Both HIS fusions resulted in fully viable and healthy strains indistinguishable from wild type. We chose to use the HIS10 fusion because it was robustly conjugated to target proteins (*Figure 1D*) and allowed better retention to the Ni-NTA resin under stringent denaturing and washing conditions (*Figure 1—figure supplement 2A, B and C*).

The Ni-NTA resin retains non-SUMOylated protein species that are recovered even in the absence of HIS10::SUMO expression (e.g., see lanes 3 and 8 in *Figure 1E and F*). These non-SUMOylated proteins are not recovered via associations with SUMO-modified proteins but rather directly due to background binding of the individual denatured protein to the Ni-affinity matrix. The recovery of these background species from the matrix is not, therefore, a good measure of the actual ratio of SUMO-modified and -unmodified species in the lysate. The unavoidable recovery of unmodified species also impairs the sensitivity of MS as a means for identifying SUMO-modified proteins. Given the very high rate of false-negative detection in SUMO MS studies, we chose to apply the lowest possible arbitrary cut off of 1 spectral count enrichment over two controls—untagged SMO-1 and depletion of HIS10::SMO-1 by smo-1(RNAi)—when generating our list of candidate SUMO conjugates. This analysis identified 977 candidate SUMO (*Supplementary file 2*; see *Supplementary file 3*, *Figure 1—figure supplement 2D*, and Discussion section for comparison with previous studies). SUMOylated MRG-1, for example, was strongly enriched by Ni-affinity chromatography as detected by MS and western blot analyses (*Supplementary file 2* and *Figure 1E and F*). Importantly, failure to detect a protein by SUMO proteomics should not be construed as evidence the protein is not modified or regulated by SUMO. For each candidate protein of interest, covalent modification by SUMO should be assessed directly with the more sensitive approach of Ni-affinity chromatography and western blotting.

PIE-1 did not pass the arbitrary cut-off we applied to our MS data (*Supplementary file 2*). We therefore carefully monitored PIE-1 SUMOylation by western blotting. To do so, we inserted a sequence encoding the FLAG epitope into the endogenous *pie-1* gene. The PIE-1::FLAG protein was expressed at similar levels in wild-type and *his10::smo-1* worms (*Figure 1E and F*), and these worms were fully viable. Ni-affinity chromatography of embryo lysates failed to enrich a modified form of PIE-1::FLAG (*Figure 1E*). However, Ni-affinity chromatography of adult lysates enriched a slowly migrating form of PIE-1::FLAG protein that was ~12 kD larger than unmodified PIE-1::FLAG

(compare lanes 14 and 19, *Figure 1F*). The modified PIE-1::FLAG band was absent when we mutated the presumptive SUMO acceptor site lysine 68 to arginine in PIE-1::FLAG (K68R) (*Figure 1F*, lane 20). Thus, lysine 68 is required for PIE-1 SUMOylation.

We were surprised to detect SUMOylated PIE-1 in adult hermaphrodites but not in embryos. Previous studies had only detected PIE-1 protein within embryonic germ cells and proximal oocytes of adult hermaphrodites (*Reese et al., 2000*; *Tenenhaus et al., 1998*). We therefore monitored PIE-1::GFP expressed from the endogenous *pie-1* locus (*Kim et al., 2014*). As expected, in embryonic germ cells, PIE-1::GFP localized to nuclei, cytoplasm, and cytoplasmic P granules (i.e., P-lineage; *Figure 1—figure supplement 3*; *Mello et al., 1996*; *Reese et al., 2000*; *Tenenhaus et al., 1998*). In the adult pachytene germline, we found that PIE-1::GFP co-localized with chromosomes (as monitored by co-localization with mCherry histone H2B) and PIE-1 protein levels appeared to gradually increase during oocyte maturation (*Figure 1G* and *Figure 1—figure supplement 3*). These findings suggest that SUMOylation of PIE-1 occurs within a heretofore unexplored zone of PIE-1 expression within the maternal germline.

## PIE-1 SUMOylation and SUMO pathway factors function together to promote fertility and embryonic development

To investigate the functional consequences of the K68R lesion, we performed a genetic complementation test with a previously described null allele, *pie-1(zu154)*. In early embryos, maternal PIE-1 promotes the germ cell fate of the P2 blastomere and prevents P2 from adopting the endoderm and mesoderm fates of its somatic sister blastomere, EMS (*Mello et al., 1992*). Hermaphrodite worms homozygous for the loss-of-function *pie-1(zu154)* allele are fertile, but 100% of their embryos arrest development with extra pharyngeal and intestinal cells (*Mello et al., 1992*). By contrast, we found that most homozygous PIE-1(K68R) animals produce viable and fertile progeny but exhibit reduced fertility and increased embryonic lethality (*Figure 2A and B*), suggesting that the K68R lesion causes a partial loss of PIE-1 function. Consistent with this idea, further lowering *pie-1* activity by placing the K68R allele over *pie-1(zu154)* dramatically enhanced the deficits in fertility (*Figure 2A*) and in embryo viability (*Figure 2B*). For example, about 11% of embryos produced by trans-heterozygotes failed to hatch (*Figure 2B*). When we examined a subset of the dead embryos by light microscopy, we found that 64% (14/22) arrested development with supernumerary intestinal cells, a hallmark of *pie-1* loss of function (*Mello et al., 1992*). Thus, PIE-1(K68R) exhibits a partial loss of PIE-1 function.

We reasoned that if the loss of SUMOylation is responsible for the hypomorphic phenotype of PIE-1(K68R) animals, then compromising the SUMO machinery might cause similar synthetic genetic interactions. To test this idea, we depleted the activity of each SUMO pathway gene by RNAi beginning at the L4 larval stage in wild-type and *pie-1(zu154)/+* heterozygotes. RNAi at the L4 stage allows the worms to produce fertilized embryos depleted of SUMO activity. When exposed to control RNAi, both wild-type and *pie-1(zu154)* heterozygous worms produce 100% viable embryos (*Figure 2C*). RNAi targeting SUMO pathway factors (*smo-1*, *gei-17*, or *ubc-9*) caused wild-type and *pie-1(zu154)* heterozygous worms to make dead embryos. Whereas only 2–20% of embryos in the wild-type background made extra intestine, about 75% of embryos made by the *pie-1(zu154)* heterozygous worms arrested with extra intestinal cells (*Figure 2C*). The synergy between *gei-17* and *pie-1(zu154)* was particularly striking: whereas *gei-17(RNAi)* in wild-type worms caused only 20% embryonic lethality, *gei-17(RNAi)* in *pie-1/+* worms caused 82% of embryos to arrest development and 94% of these produced extra intestinal cells (*Figure 2C*).

In an effort to create temperature-sensitive (ts) alleles of endogenous *ubc-9*, we used genome editing to introduce point mutations corresponding to ts alleles of yeast *ubc9* (*Figure 2D and E*; *Betting and Seufert, 1996*; *Prendergast et al., 1995*). One allele, *ubc-9(ne4446)*, which encodes a G56R amino acid substitution, resulted in worms that were viable and fertile at the permissive temperature of 15℃, but inviable when shifted to the non-permissive temperature of 25℃. Some of the temperature-sensitive phenotypes of this allele can be reversed by shifting worms back to the permissive temperature (data not shown). When shifted to 25℃ beginning at the L4 stage, *ubc-9 (ne4446[G56R])* worms developed into fertile adults that produced 100% dead embryos (n = 332) with defective body morphology but well-differentiated tissues. About 15% of *ubc-9(ne4446[G56R])* embryos arrested with extra intestinal cells, consistent with previous studies showing that in *smo-1 (RNAi)* and *ubc-9(RNAi)* embryos, the P2 blastomere expresses EMS-like cell lineage patterns, a *pie-1* mutant phenotype (*Santella et al., 2016*). Strikingly, the proportion of arrested embryos with a

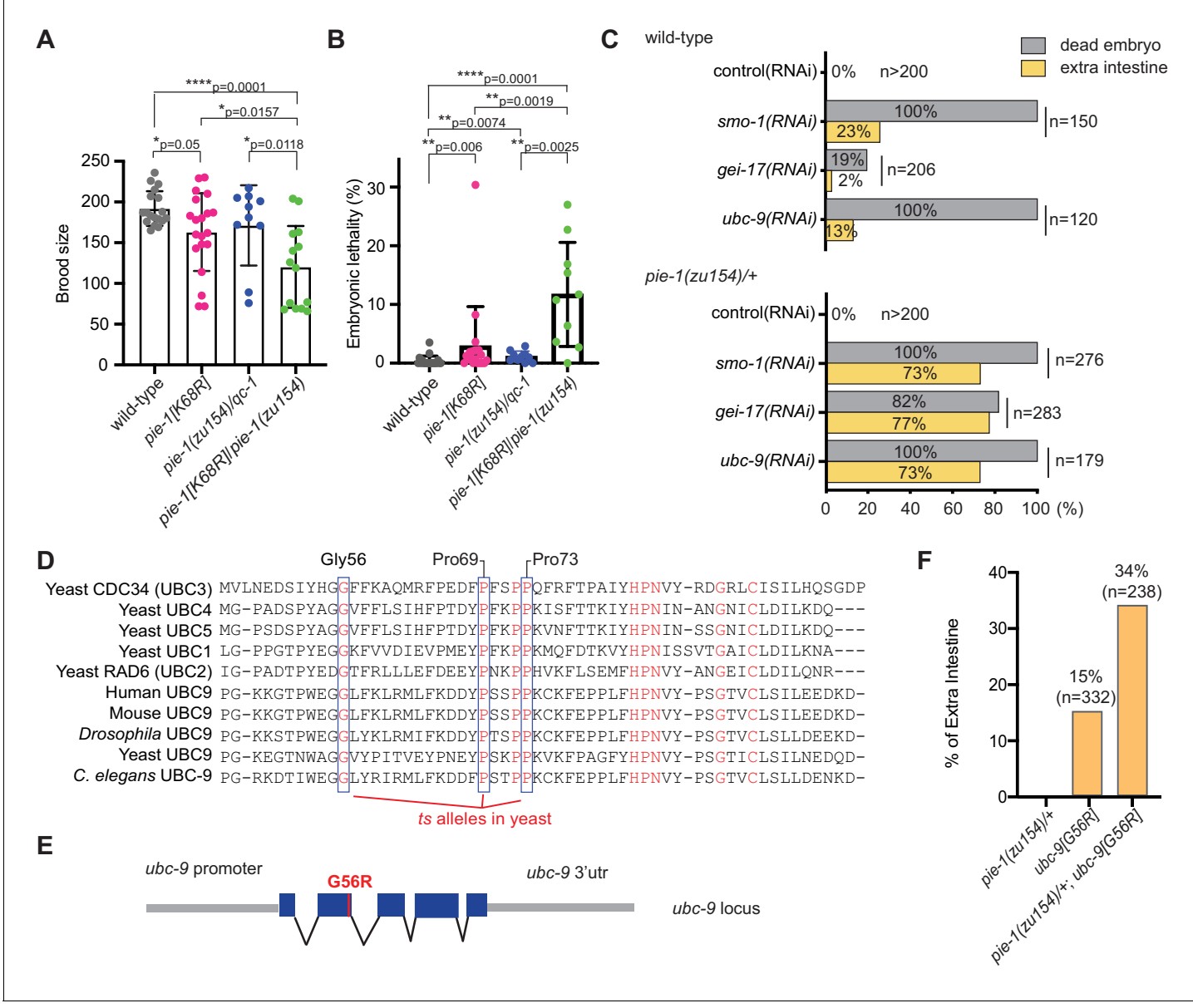

**Figure 2.** Genetic interactions between *pie-1* and SUMO pathway. (A) Brood size analysis and (B) embryonic lethality of wild-type (N2), *pie-1(ne4303 [K68R])*, *pie-1(zu154)/qC-1*, and *pie-1(ne4303[K68R])/pie-1(zu154)*. Statistical significance was determined by Wilcoxon-Mann-Whitney test: *p≤0.05; **p≤0.01; ****p≤0.0001. (C) Tests of genetic interactions between *pie-1* and SUMO pathway mutants. Bar graphs show the percentage of dead embryos (gray) and percentage of dead embryos with extra intestine (yellow) among 'n' embryos scored. (D) Partial sequence alignment of UBC enzymes, including *C. elegans* UBC-9. Residues conserved in all UBC proteins are shown in red. Temperature-sensitive (ts) alleles of yeast Cdc34 result from mutations in highly conserved residues (blue boxes). Mutating the proline resides (P69S and P73S) resulted in non-conditional lethality in *C. elegans*. A G56R mutation in *C. elegans* UBC-9 caused a ts phenotype. (E) Location of the G56R mutation introduced into the endogenous *ubc-9* gene by CRISPR genome editing. (F) Genetic interaction between *pie-1* and *ubc-9(ne4446[G56R])* allele at 25°C. Bar graphs show the percentage of embryos with extra intestine among 'n' embryos scored.

The online version of this article includes the following source data for figure 2:

**Source data 1.** Brood size and embryonic lethality.

pie-1-like phenotype was doubled when G56R homozygotes were shifted to 25°C in a heterozygous *pie-1(zu154)/+* background (*Figure 2F*), supporting the idea that SUMO promotes the activity of PIE-1 in protecting the embryonic germline.

## HDA-1 is SUMOylated in the adult

The finding that PIE-1 interacts with UBC-9, SMO-1, and GEI-17 in the yeast two-hybrid assay (*Figure 1A* and *Supplementary file 1*) raises the possibility that PIE-1 recruits the SUMO machinery to regulate its targets. If so, then one way to address the SUMO-dependent functions of PIE-1 is to search for proteins that are modified by SUMO in a PIE-1-dependent manner. To do this, we used RNAi to deplete PIE-1 from a synchronous population of young adult *his10::smo-1* animals. As controls, we used RNAi targeting *smo-1* and an empty RNAi vector (L4440). The depletion of PIE-1 and SMO-1 was monitored by western blot (*Figure 3A*). Ni-affinity chromatography followed by MS revealed 328 proteins that were reduced to a similar extent in lysates from both *pie-1(RNAi)* and *smo-1(RNAi)* animals (orange dots in *Figure 3B* and *Supplementary file 4*). Among this group of PIE-1-dependent SUMO targets, HDA-1 caught our attention because a previous study identified the nucleosome remodeling and deacetylase (NuRD) complex, including HDA-1/HDAC1 and its binding partners LET-418/Mi-2 and MEP-1, as potential targets of PIE-1 regulation (*Unhavaithaya et al., 2002*).

To directly explore the possibility that PIE-1 promotes the SUMOylation of HDA-1 and MEP-1, we used Ni-affinity chromatography followed by western blotting. To monitor SUMOylation of MEP-1, we used CRISPR to insert tandem *gfp* and *3xflag* coding sequences into the *mep-1* gene—i.e., *mep-1::gfp::tev::3xflag* (*mep-1::gtf*). For HDA-1 detection, we used a previously validated antibody (*Beurton et al., 2019*). Western blot analyses of eluates from Ni-affinity chromatography experiments revealed that slowly migrating isoforms of HDA-1 (*Figure 3C*, lanes 11 and 16) were present in adult *his10::smo-1* lysates, but absent from embryo lysates (*Figure 3C*, lane 5). By contrast, slowly migrating MEP-1 isoforms were present in both embryos and adults (*Figure 3D*, lanes 5, 11, and 16). These modified HDA-1 and MEP-1 isoforms were ~12 kD larger than unmodified HDA-1 and MEP-1, were only enriched by Ni-affinity chromatography of *his10::smo-1* lysates, and were not detected in lysates prepared from *smo-1(RNAi)* worms (*Figure 3C and D*, lane 18), suggesting that the larger isoforms result from SUMOylation.

SUMOylated HDA-1 was not detected in extracts from *pie-1(RNAi)* adults (*Figure 3C*, lane 17) and appeared reduced in adult extracts from homozygous mutant animals expressing PIE-1(K68R) (*Figure 3C*, compare lanes 11 and 12). In contrast, MEP-1 SUMOylation was unaffected by genetic perturbations of *pie-1* (*Figure 3D*, lanes 5, 12, and 17). Thus, SUMOylation of HDA-1 occurs only in adults and appears to depend—at least partly—on *pie-1* activity and PIE-1 SUMOylation.

## PIE-1 SUMOylation promotes formation of an adult germline MEP-1/HDA-1 complex

To address whether PIE-1 SUMOylation modulates the interaction of HDA-1 with other NuRD complex components, we immunoprecipitated MEP-1::GTF from early embryo and adult lysates using a GFP-binding protein (GBP) nanobody (*Rothbauer et al., 2008*) and detected LET-418 or HDA-1 by western blot. In embryo extracts, where PIE-1 and HDA-1 are not modified by SUMO, MEP-1 interacted with both LET-418 and HDA-1 (*Figure 4*, lanes 10 and 11). These robust interactions were not affected by the PIE-1(K68R) mutation (*Figure 4*, lane 12). In adult lysates, MEP-1 interacted robustly with LET-418, but only weakly with HDA-1 (*Figure 4*, lanes 22, 23, and 30). Despite a co-migrating background band present in control IPs from animals without MEP-1::GTF (see *Figure 4*, lanes 7, 8, and 9 longer exposures), this weak interaction between HDA-1 and MEP-1 was reproducibly detected, required both *smo-1* and *pie-1* activity (*Figure 4*, lanes 31 and 32), and was reduced in PIE-1(K68R) mutants (*Figure 4*, lane 24). Thus, SUMOylation of PIE-1 promotes a difficult to detect but reproducible interaction between HDA-1 and MEP-1 in the adult germline, but is not required for their much more robustly detected interaction in embryos. Given the transient nature of the SUMO modification, it is worth noting that if the interaction between HDA-1 and MEP-1 in adults requires the continued presence of HDA-1-SUMO, then our IP assays would likely underestimate the in vivo levels of the protein complex (see Discussion).

## PIE-1 suppresses histone acetylation and germline gene expression

If PIE-1 promotes the assembly of a functional HDA-1 NuRD complex in the adult germline, then we would expect global levels of acetylation to be increased in *pie-1* mutant gonads. Indeed, immunostaining revealed increased levels of histone H3 lysine 9 acetylation (H3K9Ac) in *pie-1 [K68R]* gonads

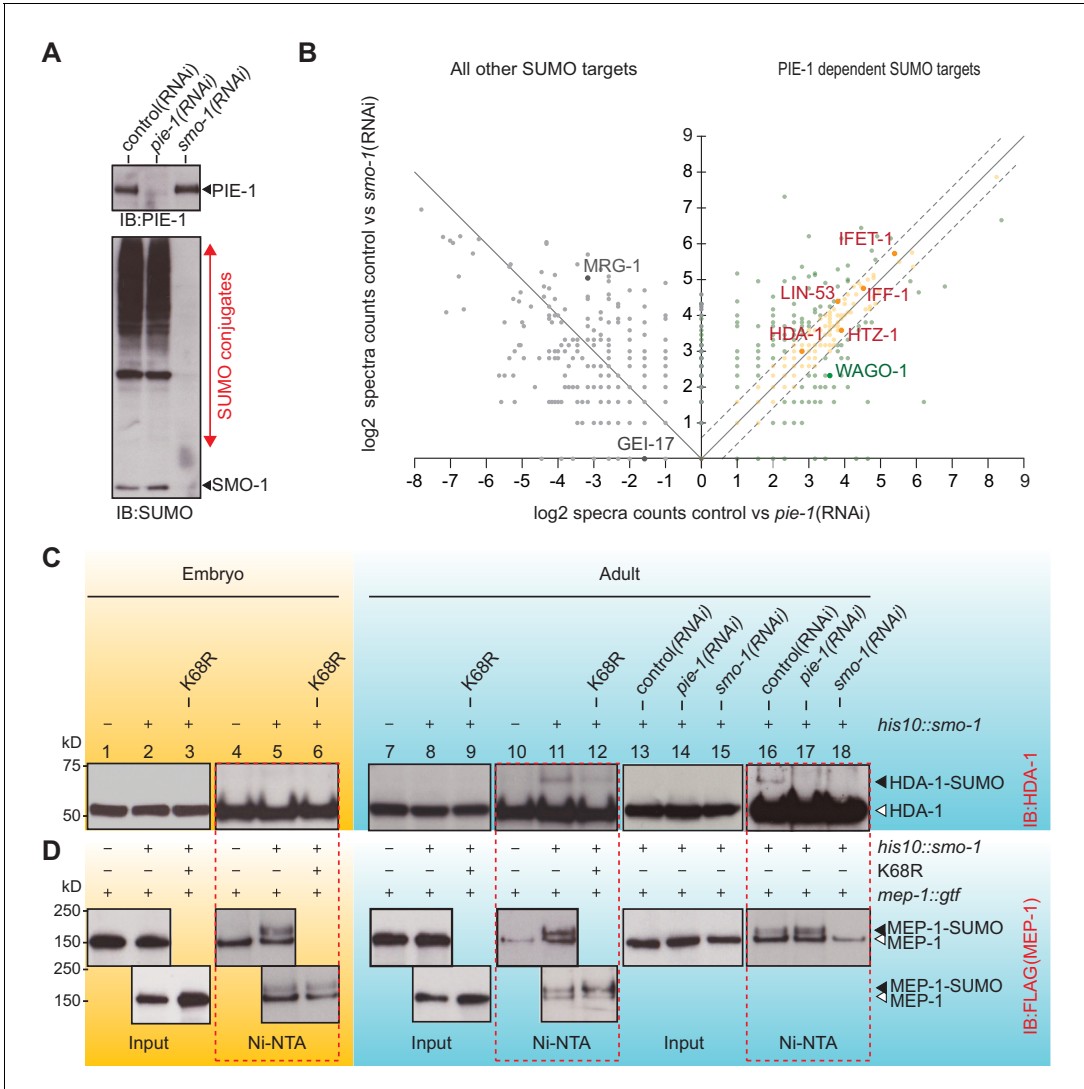

**Figure 3.** PIE-1 SUMOylation promotes HDA-1 SUMOylation in the adult germline. (A) Western blot showing relative levels of SUMOylation in HIS10::SMO-1 worms treated with control (L4440), *pie-1(RNAi)*, or *smo-1(RNAi)*. (B) Scatter plot comparing the levels of SUMOylated proteins in *pie-1(RNAi)* worms (x axis) and *smo-1(RNAi)* worms (y axis). Eluates from affinity chromatography of control, pie-1(RNAi), and smo-1(RNAi) lysates were analyzed by mass spectrometry. The log of the difference between spectral counts in control and mutant was plotted for each protein. Positive values represent proteins whose spectral counts were reduced in *pie-(RNAi)* and *smo-1(RNAi)*. Negative values on the x axis represent proteins whose spectral counts increased in *pie-1(RNAi)* compared to control. Dashed lines indicate the position of a 1.5-fold difference between the changes in *smo-1(RNAi)* and *pie-1(RNAi)* worms. A full list of PIE-1-dependent SUMO targets is provided in **Supplementary file 4**. (C and D) Western blot analyses of SUMOylated HDA-1 (C) or MEP-1 (D) enriched from embryo (yellow background) or adult (blue background) lysates of wild-type, *pie-1*, or *smo-1* mutants. Ni-NTA pull-downs are outlined by dashed red boxes. Black triangles indicate SUMOylated proteins; white triangles indicate unmodified proteins.

The online version of this article includes the following source data for figure 3:

**Source data 1.** Comparison the levels of SUMOylated proteins in pie-1(RNAi) with in smo-1(RNAi).

compared to wild-type gonads (*Figure 5A and B*), especially in the distal mitotic region and in oocytes. To more strongly deplete PIE-1 protein, we engineered an in-frame auxin-responsive *pie-1::degron::gfp* (see 'Materials and methods'). Exposing these animals to auxin from the L1 stage abolished PIE-1 expression in the adult germline (*Figure 5—figure supplement 1*) and caused a penetrant *pie-1* maternal-effect embryonic lethal phenotype. Auxin-treated adult worms showed uniformly high levels of H3K9Ac throughout the germline (*Figure 5A*). Moreover, H3K9Ac levels were higher in oocytes of auxin-treated *pie-1::degron::gfp* worms than in oocytes of *pie-1[K68R]*

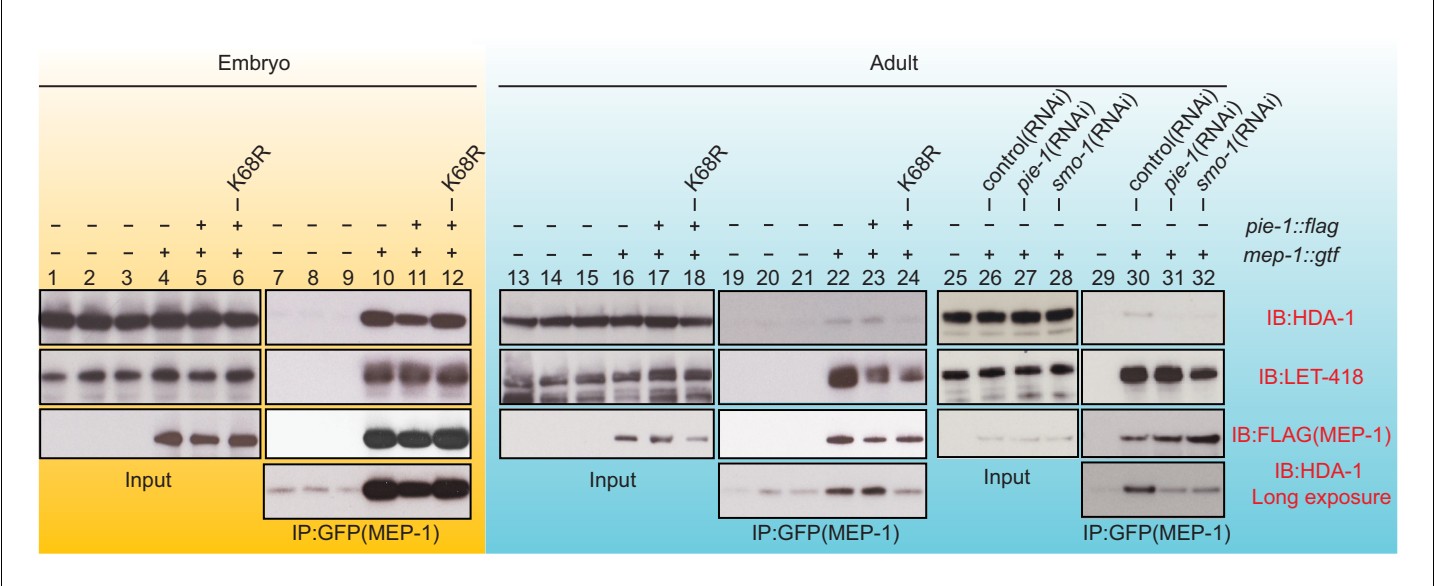

**Figure 4.** PIE-1 SUMOylation is required for the assembly of MEP-1/HDA-1 complex in the adult germline. Western blot analyses of proteins that immunoprecipitate with MEP-1::GTF from embryo (yellow background) or adult lysates (blue background) of wild-type, *pie-1*, or *smo-1* mutant worms. MEP-1::GTF was immunoprecipitated with GFP nanobody (see 'Materials and methods'). Blots were probed with HDA-1, LET-418, or anti-FLAG (MEP-1::GTF) antibodies. Longer exposure of the HDA-1 blots shows the reduced interaction of HDA-1 with MEP-1 in *pie-1* and *smo-1* mutants.

worms (*Figure 5B*). These findings suggest that PIE-1(K68R) is compromised (but not completely inactive) for its ability to promote deacetylation of H3K9 in the adult germline.

Acetylation of H3K9 is associated with active transcription (*Peterson and Laniel, 2004*), and loss of HDA-1 SUMOylation leads to increased transcriptional activity (*David et al., 2002*). We therefore sequenced mRNAs from dissected *pie-1* gonads to determine the extent to which PIE-1 regulates transcription in the adult germline. Whereas *pie-1(ne4303[K68R])* gonads exhibited mild changes in mRNA levels (*Figure 5C*), a group of 479 genes were upregulated by more than twofold in *pie-1:: degron* depleted gonads, as compared to wild type (*Figure 5D*). Upregulated protein-coding genes, included many spermatogenesis-specific genes (*Figure 5E*), suggesting that PIE-1 helps ensure a complete transition from spermatogenesis-specific gene expression to oogenesis-specific gene expression in the hermaphrodite. Five transposon families were upregulated (*Figure 5—figure supplement 2A*), including Tc5 (256-fold) and MIRAGE1 (900-fold), suggesting that PIE-1 activity also promotes transposon silencing in the adult germline. A comparison to adult gonad mRNAs upregulated in MEP-1-depleted animals and in animals homozygous for a putative HDA-1 SUMO acceptor mutant (*Kim et al., 2021*) revealed a significant overlap, especially among spermatogenesis genes (*Figure 5F*), suggesting that PIE-1 acts through the NuRD complex to regulate the majority of its adult germline targets.

## PIE-1 and GEI-17 act together to promote Piwi Argonaute-mediated silencing

In a parallel study exploring the role of SUMO and NuRD complex co-factors in piRNA-mediated silencing in the germline, we found that mutations in *smo-1* and *ubc-9* activate germline expression of a piRNA pathway reporter (*Kim et al., 2021*), but null alleles of *gei-17* do not (*Figure 6B*). We wondered if this failure might reflect a partial redundancy between GEI-17 and PIE-1 in promoting the SUMOylation of targets required for piRNA-directed transcriptional silencing. To test this, we monitored a piRNA sensor for activation in *pie-1* and *gei-17* single and double mutant strains. As a control, we monitored sensor activation in an *rde-3* mutant (*Figure 6A*), which disrupts the maintenance of silencing downstream in the piRNA pathway (*Chen et al., 2005*; *Gu et al., 2009*; *Shirayama et al., 2012*). The reporter was desilenced in 67% of first-generation *rde-3* homozygous worms and reached 100% in third-generation homozygotes (*Figure 6B*). Two different *pie-1* null alleles desilenced the piRNA sensor in 65% (15/23) and 54% (7/13) of first-generation homozygotes,

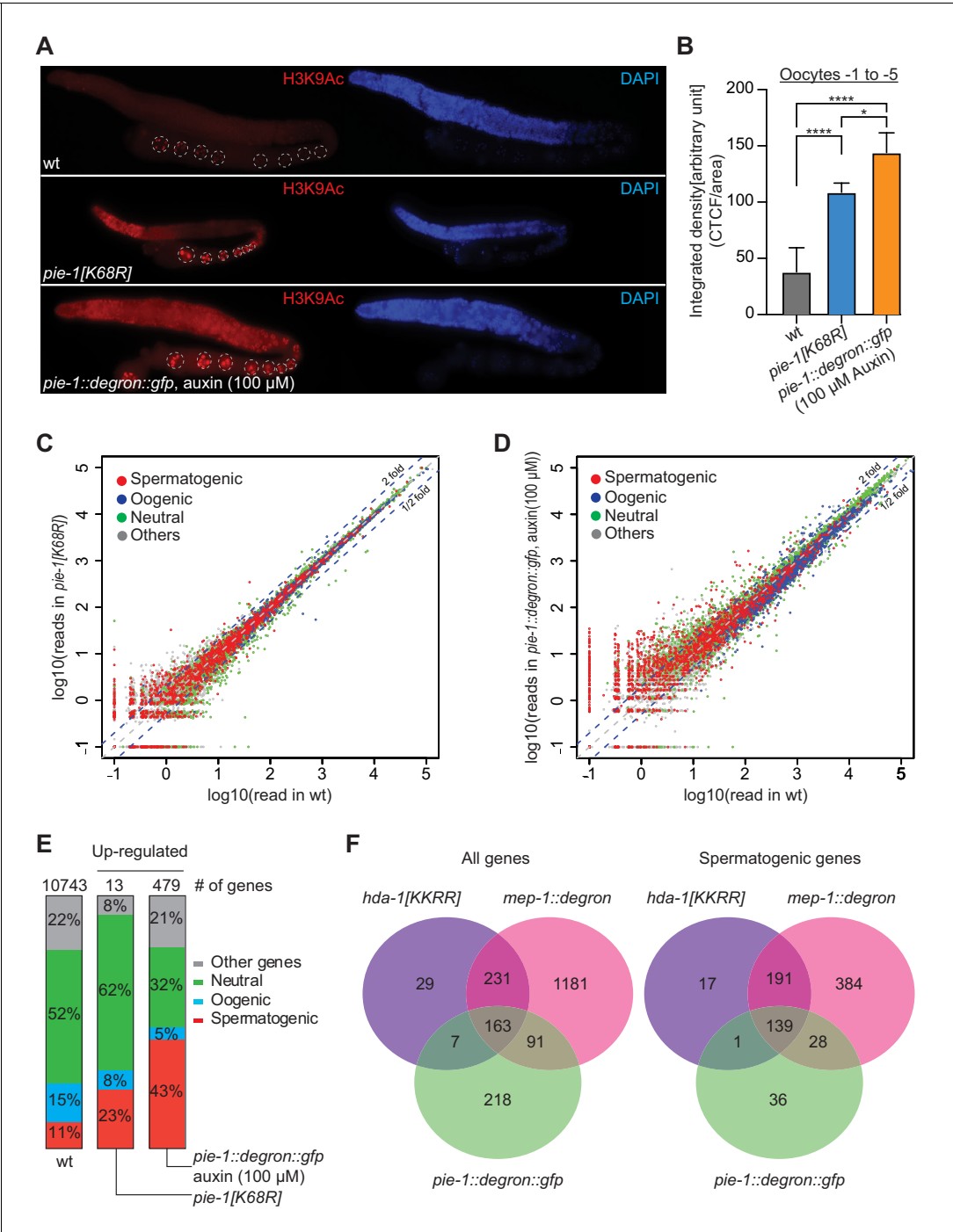

**Figure 5.** PIE-1 regulates histone H3K9Ac and spermatogenic genes in the adult germline. (**A**) Immunofluorescence micrographs of H3K9Ac and DAPI staining in adult gonad of wild-type (wt), *pie-1(ne4303[K68R])*, and *pie-1::degron::gfp* animals (100 µM auxin exposure). Oocyte nuclei are indicated with white dashed circle. (**B**) Quantification of immunofluorescence intensity in oocytes (−1 to −5). H3K9Ac signal was measured by ImageJ. The mean of the correlated total cell fluorescence (CTCF)/area ± SEM is plotted on the y axis. Significance was measured using a Tukey's test: ****p<0.0001; *p<0.05. (**C and D**) Scatter plots comparing mRNA-seq reads in (**C**) *pie-1(ne4303[K68R])* or (**D**) *pie-1::degron::gfp* to those in wt. Blue dashed lines indicate twofold increased or decreased in the mutant. Genes were categorized as spermatogenic, oogenic, neutral, or other, as defined by *Ortiz et al., 2014*. A value of 0.1 was assigned to undetected genes, thus genes with an x value of '−1' were not detected in wt. (**E**) Bar graph showing fractions of upregulated genes involved in spermatogenesis, oogenesis, neutral, or other categories. Genes expressed in wt gonads were used as a reference (10743 genes) (*Kim et al., 2021*). The number of upregulated genes in each mutant is labeled at the top. (**F**) Venn diagram showing overlap of genes upregulated in *pie-1::degron::gfp*, *hda-1[KKRR]*, and *mep-1::degron*. The *hda-1[KKRR]* and *mep-1::degron* data are from *Kim et al., 2021*. The online version of this article includes the following source data and figure supplement(s) for figure 5:

*Figure 5 continued on next page*

*Figure 5 continued*

**Source data 1.** HDAC immunostaining signal intensities.
**Figure supplement 1.** Auxin-induced depletion of PIE-1::DEGRON::GFP.
**Figure supplement 2.** Transposons upregulated in *pie-1::degron::gfp*.

but subsequent generations could not be monitored due to embryo lethality (*Figure 6B*). The *pie-1 [K68R]* mutation had a mild desilencing effect: the piRNA sensor remained silenced in the first generation, but gradually became active over the next three generations, where it was expressed in ~50% of the fourth-generation homozygotes. Two presumptive *gei-17* null alleles failed to desilence the reporter through four generations of monitoring (*Figure 6B*). While attempting to make the *gei-17; pie-1[K68R]* double mutant, we found that *gei-17/+; pie-1[K68R]* hermaphrodites are completely sterile, suggesting that *gei-17* is haploinsufficient in *pie-1[K68R]* mutants. To overcome this problem, we used CRISPR to induce homozygous *gei-17* deletion alleles directly in a piRNA sensor strain homozygous for *pie-1[K68R]*. All 18 *gei-17; pie-1[K68R]* strains recovered in this way exhibited complete first-generation desilencing of the piRNA sensor (*Figure 6B*) and were also completely sterile. In contrast, control injections into the otherwise wild-type piRNA sensor strain produced 10

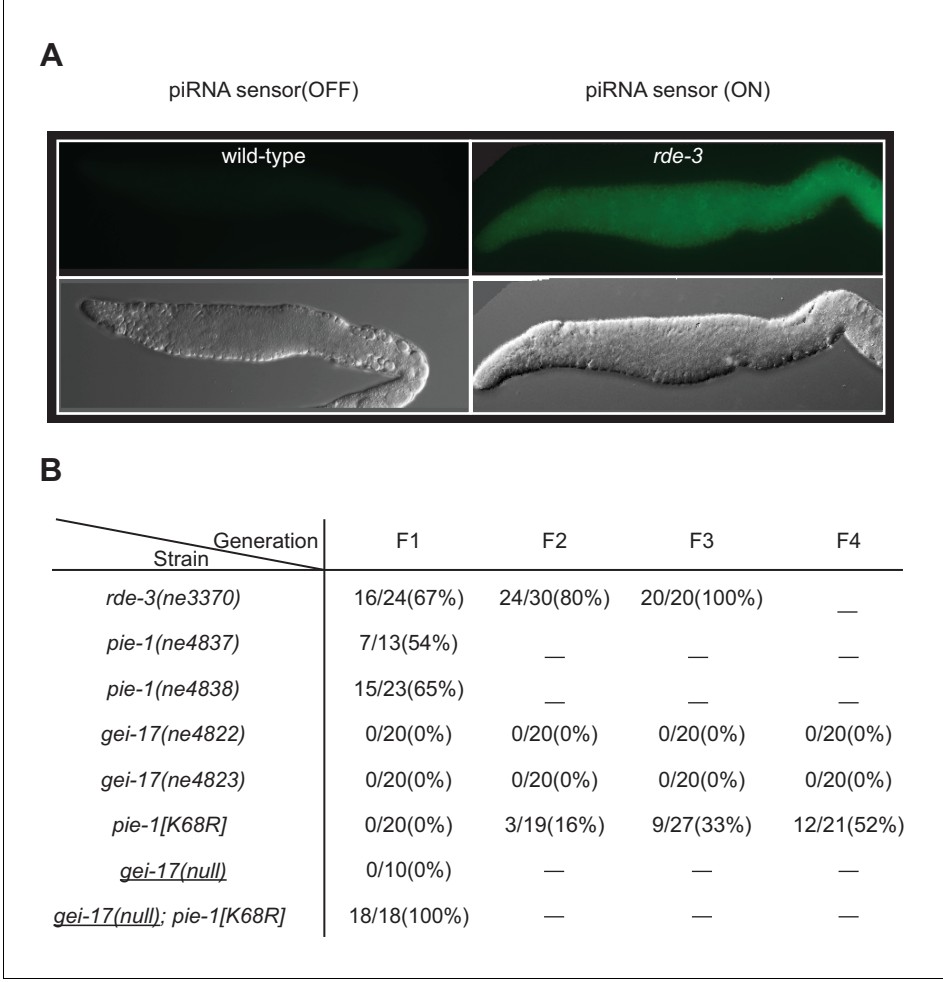

| Strain \ Generation | F1 | F2 | F3 | F4 |
|---|---|---|---|---|
| *rde-3(ne3370)* | 16/24(67%) | 24/30(80%) | 20/20(100%) | — |
| *pie-1(ne4837)* | 7/13(54%) | — | — | — |
| *pie-1(ne4838)* | 15/23(65%) | — | — | — |
| *gei-17(ne4822)* | 0/20(0%) | 0/20(0%) | 0/20(0%) | 0/20(0%) |
| *gei-17(ne4823)* | 0/20(0%) | 0/20(0%) | 0/20(0%) | 0/20(0%) |
| *pie-1[K68R]* | 0/20(0%) | 3/19(16%) | 9/27(33%) | 12/21(52%) |
| *gei-17(null)* | 0/10(0%) | — | — | — |
| *gei-17(null); pie-1[K68R]* | 18/18(100%) | — | — | — |

**Figure 6.** PIE-1 and GEI-17 function together to promote piRNA-mediated silencing. (**A**) Epifluorescence images (upper panels) of piRNA sensor expression in dissected gonads from wild-type and *rde-3(ne3370)* worms. The lower panels show differential interference contrast images of the gonads in the upper panels. (**B**) Synergistic effects of desilencing piRNA sensors in *pie-1[K68R]; gei-17* double mutants. The desilenced piRNA sensor (*gfp:: csr-1*) was scored in the indicated alleles. *gei-17(null)* alleles were generated by CRISPR editing.

homozygous *gei-17* null strains that were fertile and viable but failed to desilence in the piRNA sensor (*Figure 6B*). Thus PIE-1 and GEI-17 appear to function redundantly to promote piRNA surveillance and fertility in the adult germline.

## Discussion

The PIE-1 protein was originally identified as a maternally provided factor required to prevent early germline cells of the *C. elegans* embryo from adopting somatic differentiation programs. The PIE-1 protein was found to localize prominently to embryo germline nuclei (*Mello et al., 1996*), where it was proposed to inhibit pol II activity (*Seydoux and Dunn, 1997*; *Seydoux and Fire, 1994*), possibly through direct inhibition of the CTD kinase (*Batchelder et al., 1999*; *Ghosh and Seydoux, 2008*). Here, we have shown that PIE-1 is also expressed in the adult germline where it does not directly inhibit transcription, but rather functions along with components of the SUMO pathway to promote the hypoacetylation of germline chromatin and with the SUMO E3 homolog GEI-17 to promote Piwi Argonaute-dependent gene silencing. Although PIE-1 is not SUMOylated in the embryo, our genetic studies indicate that SUMO components, including the SUMO E3 ligase homolog GEI-17, function together with PIE-1 to preserve embryonic germline fates. PIE-1 interacts with SMO-1/SUMO and UBC-9, as well as GEI-17, by yeast two-hybrid, suggesting that PIE-1 may recruit these factors directly to its germline targets. Thus, our findings suggest how PIE-1, through its association with the SUMO pathway, may exert its remarkable dual effects as a germline determinant that controls both transcription and chromatin remodeling.

Using an affinity-based approach and mass spectrometry, we identified hundreds of *C. elegans* proteins likely to be direct SUMO conjugates. Our results complement published work identifying SUMO-regulated targets in *C. elegans* (*Drabikowski et al., 2018*; *Kaminsky et al., 2009*) and suggest that many SUMOylation substrates remain to be identified (*Figure 1—figure supplement 2D* and *Supplementary file 3*). In addition, we have shown that a HIS10::SUMO fusion expressed from the endogenous *smo-1* locus is fully functional and can be used along with mutagenesis studies to identify specific SUMO acceptor lysines in substrate proteins (see also *Kim et al., 2021*). Using these tools, we have shown that both PIE-1 and the type-1 HDAC, HDA-1, are SUMOylated in the adult germline, and that PIE-1 SUMOylation promotes the assembly of an adult germline NuRD complex.

### SUMO as an elusive modulator of protein interactions

We found that SUMO modifications to worm target proteins are rapidly and nearly completely reversed in IP sample buffer at 4°C. Consequently, SUMO-dependent protein complexes are likely to rapidly disassemble and to thus be difficult or impossible to detect in co-IP studies. Therefore, the actual in vivo interaction between MEP-1 and HDA-1 in the adult germline may be much more robust than detected here. This possibility is supported by parallel studies in which appending a conjugation-defective SUMO by translational fusion to HDA-1, HDA-1::SMO-1, dramatically increased the association of HDA-1 with MEP-1 and other components of the NuRD complex (*Kim et al., 2021*). Studies in *Drosophila* identified a complex, termed MEC, where MEP-1 resides along with Mi-2, but notably, without HDAC (*Kunert et al., 2009*). The cultured cells where this MEC complex was identified were derived from ovarian tissues and express the Piwi Argonaute (*Mugat et al., 2020*), raising the question of whether a parallel SUMO-regulated HDAC/MEC complex also functions in the *Drosophila* ovary but has been missed due to the labile nature of SUMO modifications.

In the embryo HDA-1 is not SUMOylated and yet interacts robustly with its NuRD complex co-factors. We do not know how or why the assembly of HDA-1-NuRD complex differs between embryos and adults. Perhaps, SUMOylation of HDA-1 and other NuRD complex components helps overcome an unknown barrier to co-assembly in the adult germline, enabling the proteins to associate via a mechanism bridged by SUMO (*Matunis et al., 2006*; *Psakhye and Jentsch, 2012*; *Pelisch et al., 2017*). Such a mechanism could render the assembly and function of the adult germline NuRD complex more dynamic and responsive to signals that control the conjugation and removal of SUMO.

### PIE-1 as a SUMO E3 and co-factor for the conserved SUMO-E3 GEI-17/PAIS1

Because PIE-1 appears to interact directly with UBC-9, it is possible that PIE-1 harnesses the SUMO-conjugating machinery by acting like an E3 SUMO ligase both for itself and for target proteins.

SUMOyation of PIE-1 could enhance its E3 activity and its association with UBC-9 via the non-covalent SUMO-binding site on the backside of UBC-9 (*Capili and Lima, 2007a*). The N-terminal two-thirds of PIE-1—which includes lysine 68 and both zinc finger motifs—interacts with UBC-9 in our yeast two-hybrid assays (data not shown), and the non-overlapping proline-rich C-terminal domain of PIE-1 interacts with MEP-1 (*Unhavaithaya et al., 2002*). Thus, PIE-1 may bind UBC-9 and MEP-1 simultaneously. Perhaps this ternary complex transiently associates with HDA-1 to promote its SUMOylation. HDA-1-SUMO might then associate with one or both of the putative SUMO-interacting motifs on MEP-1 (*Kim et al., 2021*).

A recent study identified the *Drosophila* GEI-17 homolog Su(var)2–10 as a bridge between a nuclear Piwi Argonaute and a chromatin remodeling complex that includes the histone methyltransferase SetDB1/Eggless (Egg) (*Ninova et al., 2020*). Our findings raise the possibility that, in addition to SetDB1, the *Drosophila* Piwi Su(var)2–10 complex may promote SUMOylation of HDAC1 to assemble a germline NuRD complex that prepares chromatin for SetDB1-dependent methylation. Perhaps in animal germlines, SUMO-dependent assembly of the NuRD complex frees it from other tasks, thereby enabling nuclear Argonautes—via associations with SUMO E3 enzymes—to recruit and assemble this vital chromatin remodeling and silencing machinery at their targets.

# Materials and methods

## Key resources table

| Reagent type (species) or resource | Designation | Source or reference | Identifiers | Additional information |
|---|---|---|---|---|
| Antibody | Mouse monoclonal anti-FLAG M2 | Sigma-Aldrich | Cat# F1804; RRID:AB_262044 | IB(1:1000) |
| Antibody | Rabbit polyclonal anti-MRG-1 | Novus Biologicals | Cat# 49130002; RRID:AB_10011724 | IB(1:1000) |
| Antibody | Rabbit polyclonal anti-HDA-1 | Novus Biologicals | Cat# 38660002; RRID:AB_10708816 | IB(1:2500) |
| Antibody | Rabbit polyclonal anti-LET-418 | Novus Biologicals | Cat# 48960002; RRID:AB_10708820 | IB(1:1000) |
| Antibody | Rat monoclonal anti-tubulin | Bio-Rad | Cat# MCA77G; RRID:AB_325003 | IB(1:2000) |
| Antibody | Mouse monoclonal anti-histone H3, acetyl K9 | Abcam | Cat# ab12179; RRID:AB_298910 | IF(1:100) |
| Antibody | Rabbit polyclonal anti-PRG-1 | *Batista et al., 2008* | N/A | IB(1:1000) |
| Antibody | Mouse monoclonal anti-SMO-1 | *Pelisch et al., 2017* | Gift from Hay Lab | IB(1:500) Freshly purified from hybridoma cell culture |
| Antibody | Mouse monoclonal anti-PIE-1(P4G5) | *Mello et al., 1996* | N/A | IB(1:100) |
| Antibody | Goat anti-mouse IgG (HRP-conjugated) | Thermo Fisher Scientific | Cat# 62–6520; RRID:AB_2533947 | IB(1:2500) |
| Antibody | Mouse anti-rabbit IgG light (HRP-conjugated) | Abcam | Cat# ab99697; RRID:AB_10673897 | IB(1:3000) |
| Antibody | Anti-rat IgG (HRP-conjugated) | Jackson ImmunoResearch Labs | Cat# 712-035-150; RRID:AB_2340638 | IB(1:5000) |
| Antibody | Goat anti-mouse IgG (H + L) Alexa Fluor 594 | Thermo Fisher Scientific | (Cat# A-11005; RRID:AB_2534073) | IF(1:1000) |
| Strain, strain background | *C. elegans* strains | This study | | *Supplementary file 5* |

*Continued on next page*

Continued

| Reagent type (species) or resource | Designation | Source or reference | Identifiers | Additional information |
|---|---|---|---|---|
| Strain, strain background | *E. coli*: strain OP50 | *Caenorhabditis* Genetics Center | WormBase: OP50 | |
| Strain, strain background | *E. coli*: strain HT115 | *Caenorhabditis* Genetics Center | WormBase: HT115 | |
| Strain, strain background | *E. coli*: Ahringer collection | Laboratory of C. Mello | N/A | |
| Peptide, recombinant protein | Ex Taq DNA polymerase | Takara | Cat# RR001C | |
| Peptide, recombinant protein | iProof high fidelity DNA polymerase | Bio-Rad | Cat#1725302 | |
| Peptide, recombinant protein | BsaI | New England Biolabs | Cat# R3535S | |
| Peptide, recombinant protein | NheI | New England Biolabs | Cat# R3131S | |
| Peptide, recombinant protein | HaeIII (screen for G56R) | New England Biolabs | Cat# R0108S | |
| Peptide, recombinant protein | Alt-R S.p. Cas9 Nuclease V3 | Integrated DNA Technologies (IDT) | Cat# 1081058 | CRISPR reagent |
| Peptide, recombinant protein | GFP-binding protein beads | Homemade | N/A | |
| Chemical compound, drug | Isopropyl-β-D-thiogalactoside | Sigma-Aldrich | Cat# 11411446001 | |
| Chemical compound, drug | Ampicillin | Sigma-Aldrich | Cat# A9518 | |
| Chemical compound, drug | Tetracycline | Sigma-Aldrich | Cat# 87128 | |
| Chemical compound, drug | Indole-3-acetic acid | Alfa Aesar | Cat# A10556 | |
| Chemical compound, drug | Tetramisole hydrochloride | Sigma-Aldrich | Cat# L9756-5G | |
| Chemical compound, drug | Paraformaldehyde 16% solution | Electron Microscopy Science | Cat# Nm15710 | |
| Chemical compound, drug | PBS | Life Technologies | Cat# AM9615 | |
| Chemical compound, drug | Tween20 | Fisher BioReagents | Cat# BP337-500 | |
| Chemical compound, drug | Bovine serum albumin | Life Technologies | Cat# AM2618 | |

*Continued on next page*

*Continued*

| Reagent type (species) or resource | Designation | Source or reference | Identifiers | Additional information |
|---|---|---|---|---|
| Chemical compound, drug | 1M HEPES, pH 7.4 | TEKnova | Cat# H1030 | |
| Chemical compound, drug | Sodium citrate dihydrate | Thermo Fisher Scientific | Cat# BP337500 | |
| Chemical compound, drug | Triton X-100 | Sigma-Aldrich | Cat# T8787-250ml | |
| Chemical compound, drug | Complete EDTA-free protease inhibitor cocktail | Roche | Cat# 11836170001 | |
| Chemical compound, drug | NP-40 | EMD Millipore | Cat# 492018 | |
| Chemical compound, drug | Tris (Base) | Avantor | Cat# 4099–06 | |
| Chemical compound, drug | Boric acid | AMRESCO | Cat# M139 | |
| Chemical compound, drug | Ethylene diaminetetraacetic acid disodium salt dihydrate | Sigma-Aldrich | Cat# E1644 | |
| Chemical compound, drug | Sodium dodecyl sulfate | Sigma-Aldrich | Cat# L3771-100G | |
| Chemical compound, drug | Sodium chloride | Genesee Scientific | Cat# 18–214 | |
| Chemical compound, drug | Magnesium chloride | Sigma-Aldrich | Cat# M8266 | |
| Chemical compound, drug | DL-dithiothreitol | Sigma-Aldrich | Cat# D0632-10G | |
| Chemical compound, drug | Potassium acetate | Fisher BioReagents | Cat# BP364-500 | |
| Chemical compound, drug | Ammonium acetate | Sigma-Aldrich | Cat# A7262 | |
| Chemical compound, drug | Deoxy-bigCHAP | Alfa Aesar | Cat# J64578-MD | |
| Chemical compound, drug | Potassium chloride | Sigma-Aldrich | Cat# P9541 | |
| Chemical compound, drug | Guanidine-HCl | Sigma-Aldrich | Cat# G3272 | |
| Chemical compound, drug | Imidazole | Sigma-Aldrich | Cat# 792527 | |

*Continued on next page*

*Continued*

| Reagent type (species) or resource | Designation | Source or reference | Identifiers | Additional information |
|---|---|---|---|---|
| Chemical compound, drug | β-Mercaptoethanol | Sigma-Aldrich | Cat# M6250 | |
| Chemical compound, drug | Sodium phosphate, dibasic | Sigma-Aldrich | Cat# S7907 | |
| Chemical compound, drug | Sodium phosphate, monobasic | Sigma-Aldrich | Cat# S0751 | |
| Chemical compound, drug | Urea | Thermo Fisher Scientific | Cat# Ac327380010 | |
| Chemical compound, drug | Trichloroacetic acid | Sigma-Aldrich | Cat# T0699 | |
| Chemical compound, drug | 1-Bromo-3-chloropropane | Sigma-Aldrich | Cat# B9673 | |
| Chemical compound, drug | TE buffer, pH 8.0 | Thermo Fisher Scientific | Cat# AM9858 | |
| Chemical compound, drug | Tris(2-carboxyethyl) phosphine hydrochloride | Sigma-Aldrich | Cat# C4706 | |
| Chemical compound, drug | Trypsin | New England Biolabs | Cat# P8101S | |
| Chemical compound, drug | TRI reagent | Sigma-Aldrich | Cat# T9424 | |
| Chemical compound, drug | Iodoacetamide | Sigma-Aldrich | Cat# I1149 | |
| Commercial assay, kit | Ni-NTA resin | Qiagen | Cat# 30210 | |
| Commercial assay, kit | SlowFade Diamond antifade Mountant with DAPI | Life Technologies | Cat# S36964 | |
| Commercial assay, kit | Quick start Bradford 1× dye reagent | Bio-Rad | Cat# 5000205 | |
| Commercial assay, kit | GlycoBlue Coprecipitant | Thermo Fisher Scientific | Cat# AM9515 | |
| Commercial assay, kit | NuPage LDS sample buffer (4×) | Thermo Fisher Scientific | Cat# NP0008 | |
| Commercial assay, kit | pCR-Blunt II-TOPO cloning kit | Thermo Fisher Scientific | Cat# K280020 | |
| Commercial assay, kit | Pierce Silver Stain Kit | Thermo Fisher Scientific | Cat# 24612 | |
| Commercial assay, kit | Lumi-Light Plus western blotting substrate | Sigma-Aldrich | Cat# 12015196001 | |
| Commercial assay, kit | Hyperfilm ECL | Thermo Fisher Scientific | Cat# 45001507 | |

*Continued on next page*

*Continued*

| Reagent type (species) or resource | Designation | Source or reference | Identifiers | Additional information |
|---|---|---|---|---|
| Commercial assay, kit | KAPA RNA HyperPrep with RiboErase (KK8560) | Roche | Cat# 08098131702 | |
| Commercial assay, kit | KAPA single-indexed adapter kit (KK8700) | Roche | Cat# 08005699001 | |
| Commercial assay, kit | Illumina NextSeq 500/550 v2.5 kit (150 cycles) | Illumina | Cat# 20024907 | |
| Recombinant DNA reagent | Peft3::cas9 vector (backbone: blunt II topo vector in this study) | *Friedland et al., 2013* | N/A | Backbone is changed to blunt II topo vector in this study |
| Recombinant DNA reagent | pRF4: injection marker, *rol-6(su1006)* | *Mello et al., 1991* | N/A | Backbone is changed to blunt II topo vector in this study |
| Recombinant DNA reagent | sgRNA plasmid | This study | | See Materials and methods; *Supplementary file 6* |
| Sequence-based reagent | gRNA and ss oligo donor sequences | This study | | *Supplementary file 6* |
| Sequence-based reagent | Alt-R CRISPR-Cas9 tracrRNA | Integrated DNA Technologies (IDT) | Cat# 1072534 | CRISPR reagent |
| Software, algorithm | GraphPad Prism version 8.2.1 | GraphPad Software | http://www.graphpad.com | |
| Software, algorithm | ImageJ | *Rueden et al., 2017* | https://imagej.net | |
| Software, algorithm | Strata 15.1 | Strata Statistical Software | http://www.strata.com | |
| Software, algorithm | Salmon | *Patro et al., 2017* | Version 1.1.0 | |
| Software, algorithm | DESeq2 | *Love et al., 2014* | Version 1.26.0 | |
| Software, algorithm | Prolucid | *Xu et al., 2006* | N/A | |
| Software, algorithm | DTASelect 2 | *Tabb et al., 2002* | N/A | |

## *C. elegans* strains and genetics

Strains and alleles used in this study were listed in *Supplementary file 5*. Worms were, unless otherwise stated, cultured at 20°C on NGM plates seeded with OP50 *Escherichia coli*, and genetic analyses were performed essentially as described (*Brenner, 1974*).

## CRISPR/Cas9 genome editing

The Co-CRISPR strategy (*Kim et al., 2014*) using *unc-22* sgRNA as a co-CRISPR marker was used to enrich HR events for tagging a gene of interest with the non-visualizing epitope (*6xhis* and *10xhis*) or introduction of a point mutation (G56R). To screen for insertions of *6xhis* and *10xhis*, we used two-round PCR: the first PCR was performed with primers (F: cctcaaaaaccaagcgaaaacc R: ccggctgctatttcattgat), and 1 µl of the first PCR product was used as a template for the second PCR with primers (F:gagactcccgctataaacga R:ctcaagcaggcgacaacgcc). To detect *6xhis or 10xhis* knock-ins, the final products were run either on 2% Tris/borate/EDTA (TBE) gel or 10% PAGE gel. *The ubc-9(G56R)*

mutation introduced an *Hae*III restriction fragment length polymorphisms (RFLP) that was used to screen for G56R conversion in PCR products (F: cattacatggcaagtcggg, R: cgttgccgcatacagaaatag). For visualization of either GFP tag, F1 rollers were mounted under coverslips on 2% agarose pads to directly screen for GFP expressing animals as described previously (*Kim et al., 2014*). *3xflag* knock-ins to *pie-1(ne4303)* were screened by PCR using previously reported primers (*Kim et al., 2014*).

sgRNA construct: Previously generated *pie-1* sgRNA plasmid (*Kim et al., 2014*) was used for *pie-1::degron::gfp* and *pie-1(K68R)::flag*. Others were constructed by ligating annealed sgRNA oligonucleotides to *Bsa*I-cut pRB1017 (*Arribere et al., 2014*) and sgRNA sequences are listed in *Supplementary file 6*.

Donor template: Unless otherwise stated, a silent mutation to disrupt the PAM site in each HR donor was introduced by PCR sewing.

*ubc-9(G56R)*: For the *ubc-9(G56R)* donor construct, a *ubc-9* fragment was amplified with primers (F: cattacatggcaagtcggg, R: gacgactaccacgaagcaagc) and this fragment was cloned into the Blunt II-TOPO vector (Thermo Fisher Scientific, K2800-20). To introduce the point mutation (G56R) and mutate the seeding region, overlap extension PCR was used.

*6xhis::smo-1/10xhis::smo-1*: Using PCR sewing, either *6xhis* (caccatcaccaccatcac) or *10xhis* fragment (caccatcaccatcaccatcaccaccatcac) was introduced immediately after the start codon in the previously generated *smo-1* fragment (*Kim et al., 2014*). The resulting PCR product was cloned into the Blunt II-TOPO vector. Tagging with *his* tag on the N-terminus of *smo-1* disrupted the PAM site.

*mep-1::gfp::tev::3xflag*: For the *mep-1::gfp::tev::3xflag* donor construct, a *mep-1* fragment was amplified with primers (F1:gaaattcgctggcagtttct R1: ctgcaacttcgatcaatcga) from N2 genomic DNA and inserted into the pCR-Blunt II-TOPO vector. Overlap extension PCR was used to introduce a *Nhe*I site immediately before the stop codon in this *mep-1* fragment. The *Nhe*I site was used to insert the *gfp::tev::3xflag* coding sequence.

*pie-1(K68R)::3xflag* and *pie-1::degron::gfp*: A previously generated donor plasmid (*Kim et al., 2014*) was used for *pie-1(K68R)::3xflag*. For *pie-1::degron::gfp* donor construct, a degron coding sequence with flanking *Nhe*I sites was amplified from pLZ29 plasmid (*Zhang et al., 2015*) and cloned the *Nhe*I fragment into a unique *Nhe*I sitee in the *pie*-1::gfp plasmid (*Kim et al., 2014*).

Other mutant alleles of *pie-1*, *gei-17*, and *rde-3* were generated by editing using Cas9 ribonucleoprotein (*Dokshin et al., 2018*). Guide RNA sequences and ssOligo donor sequence are listed in *Supplementary file 6*.

## Yeast two-hybrid analysis

The yeast two-hybrid screen was performed by Hybrigenics Services (Paris, France, http://www.hybrigenics-services.com). The coding sequence for amino acids 2–335 of *C. elegans pie-1* (NM_001268237.1) was amplified by PCR from N2 cDNA and cloned into pB66 downstream of the Gal4 DNA-binding domain sequence. The construct was sequenced and used as a bait to screen a random-primed *C. elegans* mixed-stage cDNA library in the pP6 plasmid backbone (which encodes the Gal4 activation domain). The library was screened using a mating approach with YHGX13 (Y187 ade2-101::loxp-kanMX-loxP, matα) and CG1945 (mata) yeast strains as previously described (*Fromont-Racine et al., 1997*). Five-million colonies—i.e., five times the complexity of the library—were screened. 153 His+ colonies were selected on medium lacking tryptophan, leucine, and histidine, and supplemented with 0.5 mM 3-aminotriazole to prevent bait autoactivation. The prey fragments of the positive clones were amplified by PCR and sequenced at their 5' and 3' junctions to identify the corresponding interacting proteins in the GenBank database (NCBI) using a fully automated procedure. A confidence score (predicted biological score) was attributed to each interaction as previously described (*Formstecher et al., 2005*).

## Auxin treatment

The *pie-1::degron::gfp* sequence was inserted into the endogenous *pie-1* gene in CA1199 (*unc-119 (ed3); ieSi38 [Psun-1::TIR1::mRuby::sun-1 3'UTR, cb-unc-119(+)] IV*) (*Zhang et al., 2015*) by CRISPR/Cas9-mediated genome editing. The degron-tagged L1 larvae worms were plated on NGM plates containing 100 µM auxin indole-3-acetic acid (Alfa Aesar, A10556). Worms were collected for gonad dissection when the population reached the adult stage.

## RNAi

RNAi was performed by feeding worms *E. coli* HT115 (DE3) transformed with control (empty) vector or a gene-targeting construct from the *C. elegans* RNAi collection (*Kamath and Ahringer, 2003*). For the genetic analysis, L4 larvae were placed on NGM plates containing 1 mM isopropyl β-D-thio-galactoside (IPTG) and 100 µg/ml ampicillin seeded with control or RNAi bacteria. After 24 hr, adult worms were transferred to fresh RNAi plates and allowed to lay eggs overnight. The following day, adults were removed from the test plates, and unhatched eggs were analyzed 12 hr later. For bio-chemical assays, ~200,000 synchronous L4 worms were placed on RNAi plates containing 0.4 mM IPTG, 100 µg/ml ampicillin, and 12.5 µg/ml tetracycline.

## Immunofluorescence

Gonads were dissected on a glass slide (Thermo Fisher Scientific, 1256820) in egg buffer (25 mM HEPES pH 7.5, 118 mM NaCl, 48 mM KCl, 2 mM $CaCl_2$, 2 mM $MgCl_2$) containing 0.2 mM tetramisole (Sigma-Aldrich, L9756). The samples were transferred into a Slickseal microcentrifuge tube (National Scientific, CA170S-BP) and fixed with 2% paraformaldehyde (Electron Microscopy Science, Nm15710) in phosphate-buffered saline (PBS) pH 7.4 for 10 min, and –20°C cold 100% methanol for 5 min. Fixed samples were washed four times with PBST (PBS containing 0.1% Tween20) containing 0.1% bovine serum albumin (BSA), blocked in PBST containing 1% BSA for 1 hr, and then incubated overnight at 4°C with anti-histone H3 acetyl-K9 (Abcam, ab12179) diluted 1:100 in PBST containing 1% BSA. After four washes with PBST containing 0.1% BSA, samples were incubated for 2 hr at room temperature with goat anti-mouse IgG (H + L) Alexa Fluor 594 (Thermo Fisher Scientific, A-11005) diluted 1:1000 in PBST containing 1% BSA. Samples were washed four times with PBST containing 0.1% BSA, transferred to poly-L-lysine coated slides (LabScientific, 7799) and mounted with 10 µl of SlowFade Diamond Antifade Mountant with DAPI (Life Technologies, S36964).

## Microscopy

For live imaging, worms and embryos were mounted in M9 on a 2% agarose pad with or without 1 mM tetramisole. Epifluorescence and differential interference contrast microscopy were performed using an Axio Imager M2 Microscope (Zeiss). Images were captured with an ORCA-ER digital camera (Hamamatsu) and processed using Axiovision software (Zeiss). Confocal images were acquired using a Zeiss Axiover 200 M microscope equipped with a Yokogawa CSU21 spinning disk confocal scan head and custom laser launch and relay optics (Solamere Technology Group). Stacks of images were taken and analyzed using ImageJ.

## Immunoprecipitation

Either synchronous adult worms (~200,000) or early embryos (from 10 plates of 100,000 adults) were collected, washed three times with M9 buffer, suspended in lysis buffer composed of 20 mM HEPES (pH 7.5), 125 mM sodium citrate, 0.1% (v/v) Tween 20, 0.5% (v/v) Triton X-100, 2 mM $MgCl_2$, 1 mM DTT, and a Mini Protease Inhibitor Cocktail Tablet (Roche), and homogenized in a FastPrep-24 benchtop homogenizer (MP Biomedicals). Worm or embryo lysates were centrifuged twice at 14,000 × *g* for 30 min at 4°C, and supernatants were incubated with GBP beads for 1.5 hr at 4°C on a rotat-ing shaker. The beads were washed three times with IP buffer containing protease inhibitor for 5 min each wash, and then washed twice with high-salt wash buffer (50 mM HEPES pH 7.5, 500 mM KCl, 0.05% NP40, 0.5 mM DTT, and protease inhibitor). Immune complexes were eluted with elution buffer (50 mM Tris-HCl pH 8.0, 1× SDS) for 10 min at 95°C.

## Affinity chromatography of histidine-tagged SUMO

Synchronous adult worms (~200,000) or 500 µl of packed embryos (from ~1,000,000 synchronous adult worms) were homogenized in lysis buffer at pH 8.0 (6 M guanidine-HCl, 100 mM $Na_2HPO_4$/$NaH_2PO_4$ pH 8.0, and 10 mM Tris-HCl pH 8.0) using a FastPrep-24 benchtop homogenizer (MP Bio-medicals). Lysates were cleared by centrifugation at 14,000 × *g* for 30 min at 4°C and quantified using the Quick Start Bradford 1 × Dye Reagent (BioRad, 5000205). Ni-NTA resin was washed three times with lysis buffer containing 20 mM imidazole pH 8.0 and 5 mM β-mercaptoethanol while sam-ples were prepared. To equalized samples, we added imidazole pH 8.0 to 20 mM and β-mercaptoe-thanol to 5 mM, and then the samples were incubated with 100 µl of pre-cleared 50% slurry of Ni-

NTA resin (Qiagen, 30210) for 2–3 hr at room temperature on a rotating shaker. Ni-NTA resin was washed in 1 ml aliquots of the following series of buffers: Buffer 1 pH 8.0 (6 M guanidine-HCl, 100 mM $Na_2HPO_4/NaH_2PO_4$ pH 8.0, 10 mM Tris-HCl pH 8.0, 10 mM imidazole pH 8.0, 5 mM β-mercaptoethanol, and 0.1% Triton X-100), Buffer 2 pH 8.0 (8 M urea, 100 mM $Na_2HPO_4/NaH_2PO_4$ pH 8.0, 10 mM Tris-HCl pH 8.0, 10 mM imidazole pH 8.0, 5 mM β-mercaptoethanol, and 0.1% Triton X-100), Buffer 3 pH 7.0 (8 M urea, 100 mM $Na_2HPO_4/NaH_2PO_4$ pH 7.0, 10 mM Tris-HCl pH 7.0, 10 mM imidazole pH 7.0, 5 mM β-mercaptoethanol, and 0.1% Triton X-100), Buffer 4 pH 6.3 (8 M urea, 100 mM $Na_2HPO_4/NaH_2PO_4$ pH 6.3, 10 mM Tris-HCl pH 6.3, 10 mM imidazole pH 6.3, 5 mM β-mercaptoethanol, and 0.1% Triton X-100), Buffer 5 pH 6.3 (8 M urea, 100 mM $Na_2HPO_4/NaH_2PO_4$ pH 6.3, 10 mM Tris-HCl pH 6.3, and 5 mM β-mercaptoethanol). Urea was used to denature proteins. Triton-X-100, the non-ionic detergent, was used to reduce nonspecific hydrophobic interactions. Imidazole (10 mM) was used to increase the stringency of the wash by reducing nonspecific protein binding to the resin. The use of wash buffers with gradually decreasing pH (pH 8-6.3) also reduced nonspecific binding of proteins by protonating the neutral histidine and thereby removing the weakly bound proteins that may contain tandem repeats of the histidine. SUMOylated proteins were eluted with elution buffer pH 7.0 (7 M urea, 100 mM $Na_2HPO_4/NaH_2PO_4$ pH 7.0, 10 mM Tris-HCl pH 7.0, and 500 mM imidazole pH 7.0). For western blotting, input samples containing guanidine-HCl were diluted with $H_2O$ (1:6) and then purified by trichloroacetic acid (TCA) precipitation: an equal volume of 10% TCA was added to diluted samples, which were then incubated on ice for 20 min and centrifuged for 20 min at 4°C; the obtained pellet was washed with 100 µl of ice-cold ethanol and then resuspended in Tris-HCl buffer pH 8.0.

## Western blot analysis

NuPAGE LDS sample buffer (4×) (Thermo Fisher Scientific, NP0008) was added to samples, which were then loaded on precast NuPAGE Novex 4–12% Bis-Tris protein gel (Life Technologies, NP0321BOX) and transferred onto a polyvinylidene difluoride membrane (Bio-Rad Laboratories, 1704157) using Mini Trans-Blot cells (Bio-Rad Laboratories, 1703930) at 80 V for 2.2 hr, at 4°C. Membranes were blocked with 5% (w/v) skim milk in PBST and probed with primary antibodies: 1:1000 anti-FLAG (Sigma-Aldrich, F1804); 1:1000 anti-MRG-1 (Novus Biologicals, 49130002); 1:2500 anti-HDA-1 (Novus Biologicals, 38660002); 1:1000 anti-LET-418 (Novus Biologicals, 48960002); 1:1000anti-SMO-1 (purified from Hybridoma cell cultures; the Hybridoma cell line was a gift from Ronald T. Hay, University of Dundee) (*Pelisch et al., 2017*); 1:2000 anti-tubulin (Bio-Rad, MCA77G); 1:100 anti-PIE-1(P4G5); or 1:1000 anti-PRG-1. Antibody binding was detected with secondary antibodies: 1:2500 goat anti-mouse (Thermo Fisher Scientific, 62–6520); 1:3000 mouse anti-rabbit (Abcam, ab99697); 1:5000 anti-rat (Jackson ImmunoResearch Labs, 712-035-150).

## RNA-seq

Germline RNA was extracted from 100 dissected gonads using TRI reagent (Sigma-Aldrich, T9424). Total RNA (500 ng per replica) was used for library construction using KAPA RNA HyperPrep with RiboErase (Kapa Biosystems, KK8560) and KAPA single-indexed adapter (Kapa Biosystems, KK8700) for Illumina platforms. Paired-end sequencing was performed on a NextSeq 500 Sequencer with Illumina NextSeq 500/550 high output kit v2.5 (150 cycles) (Illumina, 20024907). Salmon was used to map and quantify the reads against the worm database WS268, and its output files were imported to DESeq2 in R. Differentially expressed genes were defined as twofold change and adjusted p-value less than 0.05 (*Supplementary file 7*). The scatter plots were generated by the plot function in R. Comparisons between repeats of each sample are in *Figure 5—figure supplement 2B*.

## SUMO proteomics

HIS10::SMO-1 samples (or control samples) precipitated from adult lysates were solubilized in 8 M urea, 100 mM Tris (pH 8.0), reduced in 5 mM TCEP for 20 min, alkylated with 10 mM iodoacetamide for 15 min, and digested with trypsin. Each fraction was analyzed on a Q Exactive HF mass spectrometer (Thermo Fisher Scientific) coupled to a nano UHPLC Easy-nLC 1000 via a nano-electrospray ion source. Peptides were separated on a home-packed capillary reverse phase column (75 µm internal diameter × 15 cm of 1.8 µm, 120 Å UHPLC-XB-C18 resin) with a 110 min gradient of A and B buffers (buffer A, 0.1% formic acid; buffer B, 100% ACN/0.1% formic acid). A lock mass of

445.120025 m/z was used for internal calibration. Electrospray ionization was carried out at 2.0 kV, with the heated capillary temperature set to 275℃. Full-scan mass spectra were obtained in the positive-ion mode over the m/z range of 300–2000 at a resolution of 120,000. MS/MS spectra were acquired in the Orbitrap for the 15 most abundant multiply charged species in the full-scan spectrum having signal intensities of >1 × 10$^{-5}$ NL at a resolution of 15,000. Dynamic exclusion was set such that MS/MS was acquired only once for each species over a period of 30 s. The MS data was searched against the concatenated forward and reversed *C. elegans* protein database (WS233) using by Prolucid (*Xu et al., 2006*) and DTASelect 2 (*Tabb et al., 2002*) (≤1% false discovery rate at the peptide level).

## Acknowledgements

We thank members of Mello and Ambros labs for discussions. CCM is a Howard Hughes Medical Institute Investigator.

## Additional information

### Funding

| Funder | Grant reference number | Author |
| --- | --- | --- |
| Howard Hughes Medical Institute | | Craig C Mello |
| NIH Office of the Director | GM58800 | Craig C Mello |

The funders had no role in study design, data collection and interpretation, or the decision to submit the work for publication.

### Author contributions

Heesun Kim, Conceptualization, Data curation, Formal analysis, Validation, Investigation, Visualization, Methodology, Writing - original draft, Writing - review and editing; Yue-He Ding, Data curation, Formal analysis, Validation, Investigation, Visualization, Writing - original draft, Writing - review and editing; Shan Lu, Formal analysis, Investigation; Mei-Qing Zuo, Investigation; Wendy Tan, Validation, Investigation; Darryl Conte Jr, Conceptualization, Writing - review and editing; Meng-Qiu Dong, Supervision, Investigation; Craig C Mello, Conceptualization, Supervision, Investigation, Writing - original draft, Writing - review and editing

### Author ORCIDs

Heesun Kim  https://orcid.org/0000-0001-7643-4516
Darryl Conte Jr  https://orcid.org/0000-0002-1137-8901
Meng-Qiu Dong  http://orcid.org/0000-0002-6094-1182
Craig C Mello  https://orcid.org/0000-0001-9176-6551

### Decision letter and Author response

Decision letter https://doi.org/10.7554/eLife.63300.sa1
Author response https://doi.org/10.7554/eLife.63300.sa2

## Additional files

### Supplementary files

• Supplementary file 1. List of PIE-1 interactors identified in the yeast two-hybrid screen.

• Supplementary file 2. List of SUMO-conjugated worm proteins identified by affinity chromatography and mass spectrometry.

• Supplementary file 3. List of *C. elegans* SUMO targets also identified by *Kaminsky et al., 2009* or by *Drabikowski et al., 2018*.

• Supplementary file 4. List of PIE-1-dependent SUMO targets.

- Supplementary file 5. Strains and alleles used in this study.
- Supplementary file 6. sgRNA sequences for CRISPR.
- Supplementary file 7. RNA-seq data from dissected gonads of wild type, *pie-1(ne4303[K68R])*, and *pie-1::degron::gfp*.
- Transparent reporting form

## Data availability

RNA sequencing data have been deposited in NCBI under accession codes SRR12454798 to 12454800 and are included in Supplementary file. All data generated or analyzed during this study are included in the manuscript and supplementary files.

The following dataset was generated:

| Author(s) | Year | Dataset title | Dataset URL | Database and Identifier |
|-----------|------|---------------|-------------|-------------------------|
| Mello CC | 2021 | Gonad mRNA sequencing of pie-1 mutant in C. elegans | https://www.ncbi.nlm.nih.gov/bioproject/PRJNA657260 | NCBI BioProject, PRJNA657260 |

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
