## [Decision Letter]

**Acceptance summary:**

These studies extend the findings in the accompanying paper, which showed that sumoylation of the HDA-1 type histone deacetylase plays a role in establishing transcriptional silencing of piRNA-regulated genes in *C. elegans* by promoting local histone deacetylation. Here, the authors provide genetic evidence that the PIE-1 zinc finger protein is important for silencing of piRNA-regulated genes, and show that sumoylation of lysine 68 in the PIE-1 zinc finger protein is required for both for its association with HDA-1 and for sumoylation of HDA-1, demonstrating that PIE-1 sumoylation plays a role in piRNA-mediated silencing of gene expression in *C. elegans*.

**Decision letter after peer review:**

Thank you for submitting your article "PIE-1 promotes SUMOylation and activation of HDAC1 during the *C. elegans* oogenesis" for consideration by *eLife*. Your article has been reviewed by 3 peer reviewers, including Tony Hunter as the Reviewing Editor and Reviewer #1, and the evaluation has been overseen by Kevin Struhl as the Senior Editor. The following individual involved in review of your submission has agreed to reveal their identity: Federico Pelisch (Reviewer #2).

The reviewers have discussed the reviews with one another and the Reviewing Editor has drafted this decision to help you prepare a revised submission.

Summary:

In this paper you describe experiments showing that PIE-1 is sumoylated at K68, and that K68 sumoylation plays a role in PIE-1 interaction with HDA-1 and its sumoylation, which leads to its activation. The reviewers found the sumoylation dependence of PIE-1 function in piRNA silencing to be of interest, but raised major issues that need to be addressed. In particular, more mechanistic insights into how sumoylation of PIE-1 at K68 enhances HDA-1 sumoylation and regulation are required

Essential revisions:

1. How sumoylation of K68 affects PIE-1 function, the nature of the SUMO E3 ligase that sumoylates PIE-1, whether sumoylated PIE-1 interacts directly with HDA-1, and, and if so, whether this requires interaction of the K68 SUMO moiety with HDA-1 (e.g. via a SIM) were not elucidated, and need to be addressed with new experimental data. For instance, does a SUMO::K68R PIE-1 fusion complement the pie-1 mutant, or is the exact site of PIE-1 sumoylation important? Also, in vitro deacetylase assays of the sort suggested in the decision letter for *eLife*-63299, are needed to determine if PIE-1 K68 sumoylation leads to increased HDA-1 enzymatic activity by comparing WT and K68R PIE-1 strains. Additional suggestions for investigating the interaction of PIE-1 with HDA-1 and UBC-9 can be found in Reviewer 1's point 1.

2. Additional supporting data, including in vitro experiments (and better controlled immunoblot exposures) are needed to prove that PIE-1 sumoylation is important for interactions between HDA-1 and PIE-1, UBC-9 and MEP-1. The current data are not convincing.

Additional points are raised in the individual reviews and could be addressed, if the authors believe that addressing them would strengthen the manuscript.

*Reviewer #1:*

The evidence that sumoylation of K68 in the PIE-1 zinc finger protein is important for HDA-1 type 1 histone deacetylase association and sumoylation seems reasonable, and, is important because as shown in the co-submitted paper HDA-1 sumoylation leads to its association with MEP-1 and LET-418/NuRD complex thus accelerating H3K9ac deacetylation, and silencing gene expression..

The evidence that PIE-1 is needed for sumoylation of HDA-1, presumably though association of PIE-1 with the UBC-9 SUMO E2, is reasonable. However, several aspects of the authors' model remain unclear, and there is an absence of biochemical assays to establish the role of sumoylated PIE-1 in HDA-1 sumoylation, and the effects of sumoylation on HDA-1 HDAC activity.

1. How sumoylation of K68 in PIE1 affects its function was not worked out. Can the deleterious effect of the K68R mutation on PIE-1 function be reversed by generating a SUMO-PIE-1 fusion, as was done for HDA-1 in the co-submitted paper? K68 maps to the N-terminal side of ZF1 in the PIE-1 protein in what appears to be an unstructured region. Does the SUMO residue play a role in the interaction of PIE-1 with HDA-1? Are the zinc fingers required for PIE-1 interaction with HDA-1 or UBC-9? No zinc finger mutations were tested. Does HDA-1 have a SIM that would allow it to interact selectively with sumoylated PIE-1? Another, possibility is that the PIE-1 SUMO moiety is important because it interacts with the non-covalent SUMO-binding site on the backside of UBC-9 (Capill and Lima, JMB 369:606, 2007), which might stabilize the interaction. The backside interaction of SUMO with UBC-9 is proposed to promote UBC-9-mediated sumoylation of target proteins with SUMO consensus sites that are directly recognized by UBC-9. In this scenario, SUMO-PIE-1 would in effect be acting as an E3 SUMO ligase for HDA-1 by serving as a recruitment "factor". In this regard, the authors could test biochemically whether recombinant PIE-1 or K68SUMO-PIE-1 stimulates sumoylation of HDA-1 by UBC-9, using recombinant WT and KKRR mutant HDA-1 as substrates. These issues deserve discussion.

2. What is the SUMO E3 ligase that sumoylates PIE-1? Is it possible that through association with UBC-9, perhaps through its zinc fingers, PIE-1 is sumoylated in cis within a PIE-1/UBC-9 complex?

3. In many places, including the title, the authors make the claim that PIE-1 promotes sumoylation and activation of HDA-1. While it is clear that PIE-1 does increase sumoylation of HDA-1, in a manner requiring K68, and that H3K9ac levels are decreased as a result, the authors do not provide any direct evidence that this process increases HDA-1 catalytic activity, as is implied in the title and elsewhere. As indicated in the review of the co-submitted paper, this would need to be established by carrying out an HDAC assay on control and sumoylated HDA-1 in vitro. Instead of enzymatic activation, it is possible that the PIE-1 interaction and HDA-1 sumoylation results in relocalization of HDA-1 within the nucleus to facilitate more efficient H3K9ac deacetylation.

*Reviewer #2:*

In their manuscript, Kim et al. address the role of PIE-1 sumoylation during *C. elegans* oogenesis. The authors favour a model in which sumoylated PIE-1 acts as a sort of E3-like factor 'enhancing' HDA-1 sumoylation. While the results are indeed very interesting, it is unclear to me whether there is enough data to support the author's model. I have list of comments, suggestions, questions, and concerns, which are listed below, which I hope will help the authors strengthen the manuscript:

Figure 1:

i. As with the accompanying manuscript, the extremely low level of SUMO modification should be factored in the model.

ii. Is sumoylation also observed in untagged pie-1? As juged by figure 3A, the authors have a very good antibody to test this.

iii. While the authors claim that PIE-1 sumoylation is not observed in embryos, that panel show a lower exposure than the corresponding one in Adult (as judged by the co-purified unmodified PIE-1::FLAG). A longer exposure and/or more loading would be helpful.

iv. Their strategy and optimisation for purification of sumoylated proteins is excellent and will be useful for future research (along with other reagents the authors developed here). Is the 10xHis::smo-1 functional? Could this be tested in vitro and/or in vivo?

v. In vitro PIE-1 sumoylation would be a desirable addition to this figure.

vi. In addition to germline PIE-1 localisation, it would be interesting to see embryos and PIE-1(K68R).

vii. MW markers are missing in the blots.

Figure 2:

i. The generation of the ubc-9 ts allele is an exceptional tool. Could the authors show SUMO conjugation levels at permissive vs restrictive temperature? Just out of curiosity, is this a fast-acting allele?

ii. The authors mention that gei-17 alleles are viable, could the authors mention any thoughts on why the tm2723 allele is lethal/sterile?

Figure 3:

i. Panel C is mentioned in the text in the wrong place. Also in C, what do the authors think about the big increase in MEP-1 sumoylation in the PIE-1(K68R) background?

ii. I have the same comment for panel D as I had for figure 1 comment III: the exposure/loading for the embryo WB seems lower, as judged by the co-purifying, unmodified HDA-1. A positive control for sumoylated protein coming from embryos would be nice.

iii. In general, the model of PIE-1 acting as a SUMO machinery recruiter should be tested with recombinant proteins. Even if compatible with some results in vivo, showing that this is a plausible mechanism in vitro would be extremely helpful and greatly support the authors' claim.

Figure 4:

i. The authors make a quantitative comparison of the HDA-1/MEP-1 interaction in the text. I think this is not correct. Even if these have been run in the same gel, this could just be a lower exposure. In this line, the HDA-1 blot in the 'Adult' IP would benefit from a longer exposure to better appreciate what seems a rather small difference between PIE-1 and PIE-1(K68R).

ii. Since there still seems to be interaction between MEP-1 and HDA-1 in the PIE-1(K68R) background, does smo-1(RNAi) or ubc-9(G56R) reduce this further?

iii. In panel B, the LET-418 blot on the right is massively overexposed.

iv. Once again, in vitro binding experiments to get some indication that the authors' model is plausible would be a great addition.

Figure 5:

i. Could the authors make some quantitation of the immunofluorescence data?

Overall, I think this manuscript proposes a very interesting model and the results support this model, although I am not convinced these are sufficient to strongly back the authors' claims. I would very much like to see a revised version with some in vitro data backing the authors' model.

*Reviewer #3:*

In the manuscript by Kim et al., show that, beyond its roles of preventing somatic differentiation in the germline of embryos, Zn-finger protein PIE-1 also functions in the adult germline, where it is both SUMOylated as well as interacts with the SUMO conjugating machinery and promotes SUMOylation of protein targets. They identify HDA-1 as a target of PIE-1-induced SUMOylation. Here too, I find the claims interesting, however data is sometimes missing or does not fully support the claims.

1. A key claim of novelty over previously proposed "glue" functions of SUMO is based on the fact that they find that temporally regulated SUMOylation of a very specific residue in a specific protein is affecting protein activity: The observation that "SUMOylation of HDA-1 only appears to regulate its functions in the adult germline" and not in the embryo together with the finding that "other co-factors such as MEP-1 are SUMOylated more broadly, these findings imply that SUMOylation in the context of these chromatin remodeling complexes, does not merely function as a SUMO-glue (Matunis et al., 2006) but rather has specificity depending on which components of the complex are modified and/or when."

I find this claim poorly supported by the data. In fact, I find that the data supports that multiple SUMOylations contribute to formation of larger complexes: The His-SUMO IP (Figure 2B) brings down far more un-SUMOylated HDA-1 than SUMOylated. This argues for the presence of large complexes with different factors being SUMOylated and many bringing down unmodified HDA-1. The chromatography experiments (Figure 3B-C) also provide hits that are in complex and not direct interactors. Finally, HDA-1 SUMOylation is indicated to regulate MEP-1 interaction with numerous factors (Figure 3D). If all these factors are in one complex, it is hard to imagine how a single SUMO residue would mediate all of these simultaneously. It is quite likely (and not tested) that loss of HDA-1 SUMOylation leads to (partial?) dissociation of a large complex, rather than loss of individual interactions with the SUMO residue of HDA-1. Unlike claimed by the authors, there is no evidence that the "activity" of HDA-1 is regulated by SUMO modification.

2. Based on loss of MEP-1/HDA-1 interaction upon pie-1 RNAi and smo-1 RNAi (Figure 4B), the authors conclude that "SUMOylation of PIE-1 promotes the interaction of HDA-1 with MEP-1 in the adult germline".

The evidence that it is PEI-1 SUMOylation that is affecting MEP-1/HDA-1 interaction is fairly weak. In fact, based on Figure 4A, MEP-1 and HDA-1 interact without expression of PIE-1, and in PIE-1 K68R (sumoylation-deficient), although due to poor labeling of the panel it is not clear whether lane 1 and 4 refer to the WT pie-1 locus without tag or lack of pie-1.

In 4B the HDA-1 band that is present in L4440 but not in pie-1 or smo-1 RNAi is very faint, and in our experience such weak signal is not linear i.e., bands can disappear or appear depending on the exposure. Importantly, according to the data, seemingly unmodified HDA-1 immunoprecipitated with MEP-1 (Figure 4B). This data contradicts the authors' claim that "These findings suggest that in the adult germline only a small fraction of the HDA-1 protein pool, likely only those molecules that are SUMOylated, can be recruited by MEP-1 for the assembly of a functional NURD complex".

Furthermore, the fact that pie-1 and smo-1 depletion eliminate the interaction between HDA-1/MEP1 doesn't mean that the SUMOylation of pie-1 specifically is required for the interaction: perhaps un-SUMOylated pie1, and SUMOylation of something else, are both necessary for the interaction. The authors show that MEP-1 is also SUMOylated (Figure 3C). When IP-ing GFP-MEP-1, they precipitate all its modified forms and associated factors. One alternative possibility for why smo-1 RNAi abolishes MEP-1/HDA-1 interaction is that MEP-1 SUMOylation is needed for interaction with HDA-1 (independently of pie-1). (On a side note, why are the authors not including MEP-1 SUMOylation in the model?)

3. On page 13 the authors write: "These findings suggest that SUMOylation of PIE-1 on K68 enhances its ability to activate HDA-1 in the adult germline" and "We have shown that PIE-1 is also expressed in the adult germline where it engages the Krüppel-type zinc finger protein MEP-1 and the SUMO-conjugating machinery and functions to promote the SUMOylation and activation of the type 1 HDAC, HDA-1 (Figure 6)". Activation of HDA-1 is misleading and was never tested. If not performing in vitro assays for HDAC activity, the authors at least need to look at whether pie loss (degron) leads to acetylation of genomic HDA-1 targets and whether it affects HDA-1 (and/or MEP-1) recruitment to these sites. This could be done by ChIP-seq of HDA-1 and H3K9ac in WT and pie-1 degron animals.

---

## [Author Response]

Essential revisions:1. How sumoylation of K68 affects PIE-1 function, the nature of the SUMO E3 ligase that sumoylates PIE-1, whether sumoylated PIE-1 interacts directly with HDA-1, and, and if so, whether this requires interaction of the K68 SUMO moiety with HDA-1 (e.g. via a SIM) were not elucidated, and need to be addressed with new experimental data. For instance, does a SUMO::K68R PIE-1 fusion complement the pie-1 mutant, or is the exact site of PIE-1 sumoylation important? Also, in vitro deacetylase assays of the sort suggested in the decision letter for eLife-63299, are needed to determine if PIE-1 K68 sumoylation leads to increased HDA-1 enzymatic activity by comparing WT and K68R PIE-1 strains. Additional suggestions for investigating the interaction of PIE-1 with HDA-1 and UBC-9 can be found in Reviewer 1's point 1.

These comments made clear to us that we had placed too much emphasis on the downstream biochemical implications of our findings. We did not mean to imply that SUMO modification of HDAC increases its enzymatic activity; this possibility had not even occurred to us and seems very unlikely. The key value of our study is in the genetic insights it provides into the role of SUMO as a factor that along with PIE-1 promotes germline cell fates in the embryo and Argonaute-mediated surveillance and the assembly of an adult germline NuRD complex. We have extensively revised the paper to clarify these strengths of our story, and to remove confusion. Each of the above critiques are addressed throughout the paper, both with new experiments and with discussion.

We changed the title to downplay HDAC as the focus.

The last sentence of the Abstract now emphasizes our genetic findings and reads: *“*Our analysis of genetic interactions between *pie-1* and sumo-pathway factors suggest that PIE-1 engages the SUMO machinery both to preserve the germline fate in the embryo and to promote Argonaute-mediated surveillance in the adult germline.”

Related to this last point: We now include a new section to the results examining synthetic genetic interactions between pie-1(K68R) and the SUMO E3 ligase homolog gei-17. These experiments show that while neither single mutant causes robust de-silencing, the double shows rapid and complete desilencing of the reporter. These experiments were difficult because they uncovered a haploinsufficiency. The new Results section relating this reads as follows:

**“**PIE-1 and GEI-17 act together to promote Piwi Argonaute-mediated silencing

In a parallel study exploring the role of SUMO and NuRD complex co-factors in piRNA-mediated silencing in the germline, we found that mutations in *smo-1* and *ubc-9* activate germline expression of a piRNA pathway reporter (Kim et al., parallel), but null alleles of *gei-17* do not (Figure 6B). […] Thus PIE-1 and GEI-17 appear to function redundantly to promote piRNA surveillance and fertility in the adult germline.”

These new findings further support our new discussion of PIE-1 as an E3-like factor that acts in parallel with GEI-17.

How sumoylation of K68 affects PIE-1 function, the nature of the SUMO E3 ligase that sumoylates PIE-1.

The new Discussion also addresses these essential issues, and the new genetic data strongly support the idea that PIE-1 functions like an E3. It should also be emphasized that our findings clearly indicate how K68 affects PIE-1 activity from a genetic perspective. This is an important point, and it is not clear to us how additional mechanistic details would change the paramount importance of the key genetic findings. Unfortunately, this protein is not soluble, and without some kind of breakthrough, we don’t see how we could possibly address how K68 affects PIE-1 binding to other proteins.

Whether sumoylated PIE-1 interacts directly with HDA-1, and, and if so, whether this requires interaction of the K68 SUMO moiety with HDA-1 (e.g. via a SIM) were not elucidated, and need to be addressed with new experimental data.

Although we cannot address these questions due to the aforementioned solubility issues, we now include more data from our 2-hybrid studies. These 2-hybrid findings are discussed as they relate to the mechanism of HDA-1 regulation by PIE-1 and SUMO in the new Discussion section below:

“Because PIE-1 appears to interact directly with UBC-9, it is possible that PIE-1 harnesses the SUMO-conjugating machinery by acting like an E3 SUMO ligase both for itself and for target proteins. […] Thus PIE-1 may bind UBC-9 and MEP-1 simultaneously. Perhaps this ternary complex transiently associates with HDA-1 to promote its SUMOylation. HDA-1-SUMO might then associate with one or both of the putative SUMO-interacting motifs on MEP-1 (Kim et al., parallel).”

2. Additional supporting data, including in vitro experiments (and better controlled immunoblot exposures) are needed to prove that PIE-1 sumoylation is important for interactions between HDA-1 and PIE-1, UBC-9 and MEP-1. The current data are not convincing.

We apologize for the poor quality of some of our blots and the labeling that made it difficult to follow the subtle changes.

We do not provide any evidence that SUMOylation of PIE-1 alters direct physical interactions between PIE-1 and other factors. Again, the insolubility of PIE-1 prevents us from doing so. Instead we used the 2-hybrid, as discussed above, to find candidate interactors. To validate the interactions with UBC-9 and SUMO, it was necessary to undertake our proteomic studies of SUMO-binding proteins, which fortunately were possible under denaturing conditions. We feel that the worm SUMO proteome identified here is of extremely high quality and represents extensive efforts on the part of our collaborators in the laboratory of Dr. Men-Qiu Dong. This information and our reagents should be of significant value to the worm community.

In addition, we believe that the identification of PIE-1 as a SUMOylated protein and lysine 68 as a residue required for PIE-1 SUMOylation provide the kind of rigorous biochemical footing for our otherwise primarily in vivo genetic study. We show that *pie-1* (genetic activity) promotes the SUMOylation of HDA-1 and also promotes the association of HDA-1 with MEP-1 and other components of the NuRD complex. We agree with the reviewers that the western blots that show this relatively weak association are underwhelming. This is why we undertook a whole parallel study to more directly explore the consequences of SUMO modifications on HDA-1.

We have carefully revised the results describing the PIE-1-dependence of HDA-1 SUMOylation to be more circumspect. We have relabeled the blots and repeated with fresh antibody that now more clearly show that HDA-1 is indeed SUMOylated, and that the levels are reduced in PIE-1(K68R) and in *pie-1* and *smo-1*RNAi.

Additional points are raised in the individual reviews and could be addressed, if the authors believe that addressing them would strengthen the manuscript.Reviewer #1:The evidence that sumoylation of K68 in the PIE-1 zinc finger protein is important for HDA-1 type 1 histone deacetylase association and sumoylation seems reasonable, and, is important because as shown in the co-submitted paper HDA-1 sumoylation leads to its association with MEP-1 and LET-418/NuRD complex thus accelerating H3K9ac deacetylation, and silencing gene expression..The evidence that PIE-1 is needed for sumoylation of HDA-1, presumably though association of PIE-1 with the UBC-9 SUMO E2, is reasonable. However, several aspects of the authors' model remain unclear, and there is an absence of biochemical assays to establish the role of sumoylated PIE-1 in HDA-1 sumoylation, and the effects of sumoylation on HDA-1 HDAC activity.1. How sumoylation of K68 in PIE1 affects its function was not worked out. Can the deleterious effect of the K68R mutation on PIE-1 function be reversed by generating a SUMO-PIE-1 fusion, as was done for HDA-1 in the co-submitted paper? K68 maps to the N-terminal side of ZF1 in the PIE-1 protein in what appears to be an unstructured region.

Again, we have tried to emphasize in the revised papers that we have addressed the function of PIE-1 SUMOylation primarily with genetic studies, a strength of our system. Indeed, all of the alleles used in this study were generated at the endogenous loci by CRISPR HDR. The reviewer makes a good suggestion to attempt rescue of PIE-1(K68R) by appending SUMO through translational fusion. We attempted this with N-terminal, C-terminal, and internal fusions of SUMO into the endogenous *pie-1*orf. Unfortunately, all of these alleles caused complete embryonic lethality or strong sterility.

Does the SUMO residue play a role in the interaction of PIE-1 with HDA-1?

Again, because PIE-1 is insoluble, we cannot examine whether or not PIE-1 directly interacts with HDA-1. We show that PIE-1 and SUMO promote the interaction between HDA-1 and MEP-1, enabling them to form an adult-stage NuRD complex. K68R caused a reduction in the association between HDA-1 and MEP-1. PIE-1 does interact directly with MEP-1 and with SUMO machinery (based on 2-hybrid data).

Are the zinc fingers required for PIE-1 interaction with HDA-1 or UBC-9? No zinc finger mutations were tested.

We now include more discussion of PIE-1 two-hybrid interactions, as mentioned above, and describe how the region including the zinc fingers is required for the UBC-9 interaction, while the C-terminal proline-rich region is required for MEP-1 interactions. Mutations that alter or remove the zinc fingers of PIE-1 are not viable, making it difficult to assess their effect on the role of PIE-1 in promoting HDA-1 SUMOylation.

Does HDA-1 have a SIM that would allow it to interact selectively with sumoylated PIE-1?

There is no known SIM in HDAC. The prediction algorithm GPS-SUMO (Zhao et al., 2014) identifies one possible “low confidence” SIM in HDA-1. We have not had time to experimentally validate this one. However, as noted above we now discuss how, by forming a ternary complex with UBC-9 and MEP-1 (both of which interact strongly with PIE-1 in the 2-hybrid), PIE-1 may be able to engage and SUMOylate HDA-1 to promote its interaction with MEP-1 which does have SIM domains predicted with “high confidence” by GPS-SUMO.

Another, possibility is that the PIE-1 SUMO moiety is important because it interacts with the non-covalent SUMO-binding site on the backside of UBC-9 (Capill and Lima, JMB 369:606, 2007), which might stabilize the interaction. The backside interaction of SUMO with UBC-9 is proposed to promote UBC-9-mediated sumoylation of target proteins with SUMO consensus sites that are directly recognized by UBC-9. In this scenario, SUMO-PIE-1 would in effect be acting as an E3 SUMO ligase for HDA-1 by serving as a recruitment "factor". In this regard, the authors could test biochemically whether recombinant PIE-1 or K68SUMO-PIE-1 stimulates sumoylation of HDA-1 by UBC-9, using recombinant WT and KKRR mutant HDA-1 as substrates. These issues deserve discussion.

Thanks for these suggestions. As noted above we now discuss these ideas.

2. What is the SUMO E3 ligase that sumoylates PIE-1? Is it possible that through association with UBC-9, perhaps through its zinc fingers, PIE-1 is sumoylated in cis within a PIE-1/UBC-9 complex?

As noted above, we now discuss this important issue of E3 activities in considerable detail.

3. In many places, including the title, the authors make the claim that PIE-1 promotes sumoylation and activation of HDA-1. While it is clear that PIE-1 does increase sumoylation of HDA-1, in a manner requiring K68, and that H3K9ac levels are decreased as a result, the authors do not provide any direct evidence that this process increases HDA-1 catalytic activity, as is implied in the title and elsewhere. As indicated in the review of the co-submitted paper, this would need to be established by carrying out an HDAC assay on control and sumoylated HDA-1 in vitro. Instead of enzymatic activation, it is possible that the PIE-1 interaction and HDA-1 sumoylation results in relocalization of HDA-1 within the nucleus to facilitate more efficient H3K9ac deacetylation.

Again, we apologize for this confusion. We never meant to imply that the enzymatic activity of HDA-1 is increased by SUMOylation. We have carefully removed this wording throughout the manuscript. We think it is far more likely that HDA-1 SUMOylation promotes hypoacetylation of H3K9 by promoting the association of HDA-1 with its adult germline NuRD co-factors (as alluded to in the last comment from the reviewer above). We would be very surprised to find that a purified HDA-1::SUMO protein by itself has greater enzymatic activity. If that were to prove true, it would no doubt require a great deal of further investigation in order to provide meaningful understanding at a structural and biochemical level. Such studies and in vitro assays would go well beyond the primarily genetic and molecular studies of the kind presented here.

Reviewer #2:In their manuscript, Kim et al. address the role of PIE-1 sumoylation during *C. elegans* oogenesis. The authors favour a model in which sumoylated PIE-1 acts as a sort of E3-like factor 'enhancing' HDA-1 sumoylation. While the results are indeed very interesting, it is unclear to me whether there is enough data to support the author's model. I have list of comments, suggestions, questions, and concerns, which are listed below, which I hope will help the authors strengthen the manuscript:Figure 1:i. As with the accompanying manuscript, the extremely low level of SUMO modification should be factored in the model.

Thank you. As noted above we have added a discussion of this important issue.

ii. Is sumoylation also observed in untagged pie-1? As juged by figure 3A, the authors have a very good antibody to test this.

Unfortunately, the antibody is a monoclonal that, by chance, was raised against the peptide that includes K68. In all of our tests this antibody fails to detect the SUMOylated isoform of the protein.

iii. While the authors claim that PIE-1 sumoylation is not observed in embryos, that panel show a lower exposure than the corresponding one in Adult (as judged by the co-purified unmodified PIE-1::FLAG). A longer exposure and/or more loading would be helpful.

We have repeated this experiment and now include a longer exposure and arrows to indicate the expected size of the PIE-1-SUMO band. There is a background band that runs slightly higher in the blot, but we think the loadings are now comparable and clearly show that PIE-1 is modified by SUMO in adults but not detectably in embryos.

iv. Their strategy and optimisation for purification of sumoylated proteins is excellent and will be useful for future research (along with other reagents the authors developed here).

We thank the reviewer for these kind words. Our collaborators have developed state of the art Mass Spec data on the worm SUMO proteome.

Is the 10xHis::smo-1 functional? Could this be tested in vitro and/or in vivo?

Yes. As we now state clearly in the paper, the HIS tag is inserted at the endogenous *smo-1* locus (the only worm SUMO gene), and the strain is fully viable and healthy. This is very important as noted above, as it enables experiments like ours that identify critical lysines in the targets so that specific genetic tests of the importance of SUMO modifications on each substrate can be analyzed.

v. In vitro PIE-1 sumoylation would be a desirable addition to this figure.

We provide in vivo genetic evidence that PIE-1 and SUMO function together in both the embryo and the adult. We have directly identified PIE-1 SUMOylation in vivo with sensitive Ni-affinity and western blot studies on worm lysates, using tags engineered at the endogenous *smo-1* and *pie-1* alleles. We have also shown quite clearly – again with direct in vivo studies – that K68 is required for SUMOylation of PIE-1. We say more on this below in response to your concluding statement.

vi. In addition to germline PIE-1 localisation, it would be interesting to see embryos and PIE-1(K68R).

We now include these images as a Supplementary figure (Figure 1—figure supplement 3).

vii. MW markers are missing in the blots.

We have added markers to indicate the MW in each blot.

Figure 2:i. The generation of the ubc-9 ts allele is an exceptional tool. Could the authors show SUMO conjugation levels at permissive vs restrictive temperature? Just out of curiosity, is this a fast-acting allele?

Thanks. Yes, this is a fast-acting and reversible ts allele, so we hope it will prove useful to the community. We are glad to make it available, although it is also very easy to CRISPR it in. Based on our genetic tests the protein is a weak hypomorph (slightly reduced function), but we have not tested its SUMO-conjugating activity by comparing substrates at the permissive and non-permissive temperature.

ii. The authors mention that gei-17 alleles are viable, could the authors mention any thoughts on why the tm2723 allele is lethal/sterile?

In our hands gei-17 null alleles are homozygous viable, but produce a mixture of ~70% sterile adults and ~2% dead embryos in each brood. We have not analyzed tm2723, but note that it is available as an unbalanced (homozygous) strain according to wormbase.

Figure 3:i. Panel C is mentioned in the text in the wrong place. Also in C, what do the authors think about the big increase in MEP-1 sumoylation in the PIE-1(K68R) background?

We have corrected the figure calls, and we have rearranged the figure to more clearly and concisely indicate the embryo and adult studies. You are right, MEP-1 SUMOylation does seem to be increased slightly in the mutant and in *pie-1(RNAi)*. However, we have not identified the SUMO acceptor lysines nor generated any functional data indicating the importance (if any) of MEP-1 SUMOylation. Investigating this modification to MEP-1 will require a future study.

ii. I have the same comment for panel D as I had for figure 1 comment III: the exposure/loading for the embryo WB seems lower, as judged by the co-purifying, unmodified HDA-1. A positive control for sumoylated protein coming from embryos would be nice.

We have adjusted the loading and include MEP-1 as a positive control for a SUMOylated protein from embryos.

iii. In general, the model of PIE-1 acting as a SUMO machinery recruiter should be tested with recombinant proteins. Even if compatible with some results in vivo, showing that this is a plausible mechanism in vitro would be extremely helpful and greatly support the authors' claim.

We thank the reviewer for this perspective. We agree that in vitro studies would be helpful in the future, especially if they help uncover factors or modifications that cause adult germline and embryo MEP-1 Mi2/LET-418 complexes to differ so dramatically in their association with HDA-1. Especially if these factors—whatever they are—turn out to be highly conserved proteins (unlike PIE-1). Until that problem is solved, even if PIE-1 could be shown to promote HDA-1 SUMOylation in vitro with purified components, it would be unlikely to modulate the association of HDA-1 with MEP-1 in vitro, as the aforementioned modifications or co-factors that suppress/promote their association are not yet in hand.

At present, our genetic arguments strongly support the idea that PIE-1 works together with SUMO components to carry out its functions. We feel like our hard fought gains in understanding and revealing a potential mechanism for PIE-1 adult germline function (by promoting HDAC NuRD complex assembly) are very significant given the strong context of the entirely in vivo genetic backdrop for these whole animal studies. We feel like this genetic in vivo aspect of the story is solid and very interesting and is already well-supported by good molecular genetic studies using endogenous alleles throughout.

Figure 4:i. The authors make a quantitative comparison of the HDA-1/MEP-1 interaction in the text. I think this is not correct. Even if these have been run in the same gel, this could just be a lower exposure. In this line, the HDA-1 blot in the 'Adult' IP would benefit from a longer exposure to better appreciate what seems a rather small difference between PIE-1 and PIE-1(K68R).

We now include a longer exposure that helps to make this difference more clear. It should be noted that the SUMO-dependence of this interaction is pursued more directly and completely in our parallel story on HDAC SUMOylation. Although it may seem weak, it is most definitely real and, we believe, important. In the revised Discussion, we suggest that this apparently weak association is likely misleading due to SUMO reversal during IP incubations.

ii. Since there still seems to be interaction between MEP-1 and HDA-1 in the PIE-1(K68R) background, does smo-1(RNAi) or ubc-9(G56R) reduce this further?

As indicated in the revised Figure 4, lanes 31 and 32, both *pie-1(RNAi) and smo-1(RNAi)* appear to reduce this interaction further.

iii. In panel B, the LET-418 blot on the right is massively overexposed.

We provide a shorter exposure, but LET-418 interacts very robustly with MEP-1 in adult lysates, which is what we were trying to illustrate. When relative exposures are adjusted to detect HDA-1, this over-exposure happens to the LET-418 western.

iv. Once again, in vitro binding experiments to get some indication that the authors' model is plausible would be a great addition.

Thanks again for this comment. We have responded to this issue of in vitro studies above. But please clarify if there are specific new in vitro experiments you have in mind. Again, the genetic arguments we make are very well supported by molecular genetic studies and by yeast two-hybrid data.

It is not clear what aspect of our model seems implausible. We do agree with the reviewer that the role of PIE-1 in promoting HDA-1 SUMOylation and its role in the regulation of the adult stage MEP-1–HDA-1 interaction remains somewhat tentative (but is addressed more deeply in the second paper). But we don’t think they are implausible. In the revised manuscript, we have de-emphasized these molecular details (taking them out of the title, for example). We think the findings that PIE-1 is SUMOylated and acts through SUMO to regulate gene expression in both the adult and embryo germlines are important and well-established aspects of our paper. As noted above, the importance of HDAC SUMOylation is addressed much more completely in the parallel submission.

Figure 5:i. Could the authors make some quantitation of the immunofluorescence data?

We now include quantification (Figure 5B).

Overall, I think this manuscript proposes a very interesting model and the results support this model, although I am not convinced these are sufficient to strongly back the authors' claims. I would very much like to see a revised version with some in vitro data backing the authors' model.

We hope the reviewer will be satisfied with the revisions we have made to emphasize the molecular genetic (rather than biochemical) focus of our work. We have emphasized and expanded on our 2-hybrid data in order to help address this need for in vitro mechanistic binding information. PIE-1 is a difficult protein in terms of solubility, both in vivo and in our limited attempts to work with it in vitro. It is also a novel protein, that beyond its tandem zinc fingers is not conserved. So detailed in vitro insights into how PIE-1 SUMOylation changes its function are unlikely to be of great interest to the readers of *eLife*. As noted earlier, a key mystery that deserves further investigation at the biochemical level is why HDA-1 fails to associate with the NuRD complex in the adult germline, in the first place. In any event, thanks for your careful consideration of our work, and we apologize that we cannot be more forthcoming with in vitro data.

Reviewer #3:In the manuscript by Kim et al., show that, beyond its roles of preventing somatic differentiation in the germline of embryos, Zn-finger protein PIE-1 also functions in the adult germline, where it is both SUMOylated as well as interacts with the SUMO conjugating machinery and promotes SUMOylation of protein targets. They identify HDA-1 as a target of PIE-1-induced SUMOylation. Here too, I find the claims interesting, however data is sometimes missing or does not fully support the claims.1. A key claim of novelty over previously proposed "glue" functions of SUMO is based on the fact that they find that temporally regulated SUMOylation of a very specific residue in a specific protein is affecting protein activity: The observation that "SUMOylation of HDA-1 only appears to regulate its functions in the adult germline" and not in the embryo together with the finding that "other co-factors such as MEP-1 are SUMOylated more broadly, these findings imply that SUMOylation in the context of these chromatin remodeling complexes, does not merely function as a SUMO-glue (Matunis et al., 2006) but rather has specificity depending on which components of the complex are modified and/or when."I find this claim poorly supported by the data. In fact, I find that the data supports that multiple SUMOylations contribute to formation of larger complexes: The His-SUMO IP (Figure 2B) brings down far more un-SUMOylated HDA-1 than SUMOylated. This argues for the presence of large complexes with different factors being SUMOylated and many bringing down unmodified HDA-1.

As we stated in the paper, the Ni-affinity enrichments are performed under denaturing conditions and therefore do not pull down complexes bridged by protein-protein interactions. This is why PIE-1 protein which is otherwise not soluble, was recovered using this method. The non-SUMOylated proteins recovered in these experiments are brought down individually (not in complexes), and presumably this is due to unavoidable background binding between the Ni resin and clusters of Histidines in these proteins. Background binding to the resin is different for every protein, and is not by itself a useful measure of the relative SUMOylation level of the protein. We now explain the procedure and emphasize the points more strongly in the revised paper. Thanks for pointing out these issues as no doubt many other readers would be unfamiliar with such details.

The chromatography experiments (Figure 3B-C) also provide hits that are in complex and not direct interactors.

The reviewer is referring to Figure 3 in the HDA-1 paper, not this paper. Again, the histograms compare the recovery of proteins by Ni-affinity chromatography under ***denaturing conditions*,** which does not capture protein complexes. We have clarified in both papers that Ni-affinity chromatography was done under denaturing conditions, and we apologize for the confusion.

Finally, HDA-1 SUMOylation is indicated to regulate MEP-1 interaction with numerous factors (Figure 3D). If all these factors are in one complex, it is hard to imagine how a single SUMO residue would mediate all of these simultaneously.

Yes, we agree; this is a striking finding. Appending SUMO by translational fusion to HDA-1 results in a dramatic increase in the co-IP between MEP-1, HDA-1, and several other factors that are known to form complexes with HDA-1. As we discuss in the paper, appending SUMO to HDA-1 seems to overcome a barrier to its interaction with MEP-1 and enables the proteins to assemble a functional NuRD complex in the adult germline.

It is quite likely (and not tested) that loss of HDA-1 SUMOylation leads to (partial?) dissociation of a large complex, rather than loss of individual interactions with the SUMO residue of HDA-1.

We don’t understand this comment, which again seems to be addressing the HDA-1 paper, not this paper. Perhaps the reviewer did not understand that we pulled down MEP-1 not HDA-1 in Figure 3D of the HDA-1 paper.

Unlike claimed by the authors, there is no evidence that the "activity" of HDA-1 is regulated by SUMO modification.

We apologize again for this misunderstanding. We agree that SUMO is unlikely to alter the HDA-1 enzymatic activity. Rather as now more clearly stated throughout the paper, we believe it alters its activity toward its histone substrates by enhancing its association with MEP-1.

2. Based on loss of MEP-1/HDA-1 interaction upon pie-1 RNAi and smo-1 RNAi (Figure 4B), the authors conclude that "SUMOylation of PIE-1 promotes the interaction of HDA-1 with MEP-1 in the adult germline".The evidence that it is PEI-1 SUMOylation that is affecting MEP-1/HDA-1 interaction is fairly weak. In fact, based on Figure 4A, MEP-1 and HDA-1 interact without expression of PIE-1, and in PIE-1 K68R (sumoylation-deficient), although due to poor labeling of the panel it is not clear whether lane 1 and 4 refer to the WT pie-1 locus without tag or lack of pie-1.

Again we apologize for the confusion and poor labeling of our figures. We now make it clear that all of the strains are wildtype for PIE-1 function with the exception of those expressing PIE-1(K68R) or exposed to *pie-1(RNAi)*. The dash was meant to indicate the absence of the tag not the protein or gene activity. In the original figure, the top panels showed results from adult lysates, and the bottom panels showed results from embryo lysates. As we note in the paper and as mentioned by the reviewer, HDA-1 and MEP-1 interact robustly in embryos regardless of the SUMOylation status of HDA-1. To make the experiments and results clearer, we now place the two conditions—Adult and Embryo—side by side and color code them. We also put the most important blots (HDA-1) on top and call out all the key lanes by number. As noted above and now emphasized throughout the paper, the interaction between MEP-1 and HDA-1 is weak in the Adult stage, and this weak interaction requires PIE-1 activity and is promoted by PIE-1 SUMOylation. We also now make a point of discussing how this apparently weak interaction may be a detection problem due to reversal of SUMO in the IP lysates.

In 4B the HDA-1 band that is present in L4440 but not in pie-1 or smo-1 RNAi is very faint, and in our experience such weak signal is not linear i.e., bands can disappear or appear depending on the exposure.

Under the conditions used in our study, a relatively very small amount of HDA-1 is recovered in the MEP-1 IPs from the adult germline compared to the embryo. We also include longer exposures now that show this better. The interaction is reproducible, and is consistently reduced in *K68R, pie-1(RNAi)*, and *smo-1(RNAi)* worms. In contrast Mi2/LET-418 interacts robustly with MEP-1 at all stages and in all genetic contexts. We intentionally exposed the blots from Adults and Embryos for equal amounts of time so that the relative amount of LET-418 blot and HDA-1 recovered in the IPs could be compared directly. Hence the over-exposure of LET-418 and under-exposure of the HDA-1. At the request of the reviewers, we have added additional figures showing longer exposures of HDA-1 in the Adult stage coIP.

Importantly, according to the data, seemingly unmodified HDA-1 immunoprecipitated with MEP-1 (Figure 4B). This data contradicts the authors' claim that "These findings suggest that in the adult germline only a small fraction of the HDA-1 protein pool, likely only those molecules that are SUMOylated, can be recruited by MEP-1 for the assembly of a functional NURD complex".

We thank the reviewer for pointing this out, and again we apologize for not explaining this finding better. SUMO modifications are transient and reversible and are often subject to reversal by de-SUMOylating machinery during incubation in IP lysates. We now address this issue directly in the Results and Discussion sections and include a figure (1D) to illustrate this rapid de-conjugation of SUMO from substrates in IP buffer.

Furthermore, the fact that pie-1 and smo-1 depletion eliminate the interaction between HDA-1/MEP1 doesn't mean that the SUMOylation of pie-1 specifically is required for the interaction: perhaps un-SUMOylated pie1, and SUMOylation of something else, are both necessary for the interaction.

We agree with the reviewer. We show that PIE-1(K68R) reduces but does not eliminate the interaction, and we speculate that PIE-1, even when not SUMOylated can promote the interaction. Our best evidence is genetic, so of course it could be indirect. However, PIE-1 does directly interact with MEP-1, and with SUMO and UBC-9, so a direct modification of HDA-1 is certainly plausible. We have better explained and de-emphasized this detail and hope this makes the paper more acceptable to the reviewer.

The authors show that MEP-1 is also SUMOylated (Figure 3C). When IP-ing GFP-MEP-1, they precipitate all its modified forms and associated factors. One alternative possibility for why smo-1 RNAi abolishes MEP-1/HDA-1 interaction is that MEP-1 SUMOylation is needed for interaction with HDA-1 (independently of pie-1). (On a side note, why are the authors not including MEP-1 SUMOylation in the model?)

MEP-1 is SUMOylated in both the embryo and the adult, as are additional components of the NuRD complex. While it’s possible that this modification is essential for its (their) functions in one or both tissues, we have not explored this yet. A careful exploration of this question will require finding and mutating the acceptor lysines.

3. On page 13 the authors write: "These findings suggest that SUMOylation of PIE-1 on K68 enhances its ability to activate HDA-1 in the adult germline" and "We have shown that PIE-1 is also expressed in the adult germline where it engages the Krüppel-type zinc finger protein MEP-1 and the SUMO-conjugating machinery and functions to promote the SUMOylation and activation of the type 1 HDAC, HDA-1 (Figure 6)". Activation of HDA-1 is misleading and was never tested.

We did not mean to imply enzymatic activity was increased. This section has been totally revised to stress the key genetic findings.

If not performing in vitro assays for HDAC activity, the authors at least need to look at whether pie loss (degron) leads to acetylation of genomic HDA-1 targets and whether it affects HDA-1 (and/or MEP-1) recruitment to these sites. This could be done by ChIP-seq of HDA-1 and H3K9ac in WT and pie-1 degron animals.

Thanks for the suggestion, it is difficult/impossible to collect enough material by dissecting gonads to be able to perform ChIP seq studies on adult germline tissue from worms. One solution is to drive a tagged transgene and perform ChIP that way. We did this in the HDA-1 paper. However, we felt that performing mRNA seq on the actual mutant alleles, directly, gives a similar read out on overlap between targets. We now include direct comparisons of up-regulated genes in dissected gonads from *hda-1([KKRR])*, *mep-1::degron*, and *pie-1::degron*, showing strong overlaps (Figure 5F).